# Treatment effect heterogeneity following type 2 diabetes treatment with GLP1-receptor agonists and SGLT2-inhibitors: a systematic review

Katherine G. Young[1,201], Eram Haider McInnes[2,201], Robert J. Massey[2,201], Anna R. Kahkoska[3], Scott J. Pilla[4], Sridharan Raghavan[5], Maggie A. Stanislawski [6], Deirdre K. Tobias[7,8], Andrew P. McGovern[1], Adem Y. Dawed[2], Angus G. Jones[1], Ewan R. Pearson [2✉], John M. Dennis [1✉] & ADA/EASD PDMI*

## Abstract

**Background** A precision medicine approach in type 2 diabetes requires the identification of clinical and biological features that are reproducibly associated with differences in clinical outcomes with specific anti-hyperglycaemic therapies. Robust evidence of such treatment effect heterogeneity could support more individualized clinical decisions on optimal type 2 diabetes therapy.

**Methods** We performed a pre-registered systematic review of meta-analysis studies, randomized control trials, and observational studies evaluating clinical and biological features associated with heterogenous treatment effects for SGLT2-inhibitor and GLP1-receptor agonist therapies, considering glycaemic, cardiovascular, and renal outcomes. After screening 5,686 studies, we included 101 studies of SGLT2-inhibitors and 75 studies of GLP1-receptor agonists in the final systematic review.

**Results** Here we show that the majority of included papers have methodological limitations precluding robust assessment of treatment effect heterogeneity. For SGLT2-inhibitors, multiple observational studies suggest lower renal function as a predictor of lesser glycaemic response, while markers of reduced insulin secretion predict lesser glycaemic response with GLP1-receptor agonists. For both therapies, multiple post-hoc analyses of randomized control trials (including trial meta-analysis) identify minimal clinically relevant treatment effect heterogeneity for cardiovascular and renal outcomes.

**Conclusions** Current evidence on treatment effect heterogeneity for SGLT2-inhibitor and GLP1-receptor agonist therapies is limited, likely reflecting the methodological limitations of published studies. Robust and appropriately powered studies are required to understand type 2 diabetes treatment effect heterogeneity and evaluate the potential for precision medicine to inform future clinical care.

## Plain language summary

This study reviews the available evidence on which patient features (such as age, sex, and blood test results) are associated with different outcomes for two recently introduced type 2 diabetes medications: SGLT2-inhibitors and GLP1-receptor agonists. Understanding what individual characteristics are associated with different response patterns may help clinical providers and people living with diabetes make more informed decisions about which type 2 diabetes treatments will work best for an individual. We focus on three outcomes: blood glucose levels (raised blood glucose is the primary symptom of diabetes and a primary aim of diabetes treatment is to lower this), heart disease, and kidney disease. We identified some potential factors that reduce effects on blood glucose levels, including poorer kidney function for SGLT2-inhibitors and lower production of the glucose-lowering hormone insulin for GLP1-receptor agonists. We did not identify clear factors that alter heart and kidney disease outcomes for either medication. Most of the studies had limitations, meaning more research is needed to fully understand the factors that influence treatment outcomes in type 2 diabetes.

A full list of author affiliations appears at the end of the paper.

Two of the most recently introduced anti-hyperglycaemic drug classes, SGLT2-inhibitors (SGLT2i) and GLP1-receptor agonists (GLP1-RA), have been shown in randomized clinical trials not only to reduce glycaemia[1] but also to lower the risk of renal and cardiovascular disease (CVD) outcomes among high-risk individuals with type 2 diabetes (T2D)[2–5]. Based on average treatment effects reported in placebo-controlled trials, current T2D clinical consensus guidelines recommend a stratified approach to treatment selection, preferentially recommending these drug classes independent of their glucose lowering effect for individuals with cardiovascular or renal comorbidity. Specifically, people with heart failure and/or chronic kidney disease are recommended to initiate SGLT2i and people with prior CVD or high risk for CVD are recommended to initiate either an SGLT2i or a GLP1-RA. In addition, these drugs are recommended as second-line glucose lowering medications to be added after metformin[6].

A limitation of the current stratified approach to SGLT2i and GLP1-RA treatment in clinical guidelines is that it is informed by selective trial recruitment strategies, and consequential accumulation of evidence of treatment benefits only for specific subgroups with or at high risk of cardiorenal disease, rather than from an understanding of how the benefits and risks of each drug class vary across the whole spectrum of T2D. A more comprehensive approach to treatment selection would require recognition of the extreme heterogeneity in the demographic, clinical, and biological features of people with T2D, and the impact of this heterogeneity on drug-specific clinical outcomes. Identification of robust and reproducible patterns of heterogenous treatment effects is plausible as, at the individual patient level, responses to the same drug treatment appear to vary greatly[7]. A greater understanding of population-wide heterogenous treatment effects and enhanced capacity to predict individual treatment responses is needed to advance towards the central goal of precision type 2 diabetes medicine—using demographic, clinical, biological, or other patient-level features to match individuals to their optimal anti-hyperglycaemic regimen as part of routine T2D clinical care.

To assess the evidence base for treatment effect heterogeneity for SGLT2i and GLP1-RA, we undertook a systematic literature review to summarize key findings from studies that specifically examined interactions between individual-level biomarkers and the effects of these drug classes on clinical outcomes. Although biomarkers may connote laboratory-based measurements in traditional contexts, herein we broadly conceptualized biomarkers as individual-level demographic, clinical, and biological features, including both laboratory measures as well as genetic and genomic markers. We focused on three categories of outcomes relevant to T2D care: (1) glycaemic response (as measured by hemoglobin A1c; HbA1c); (2) CVD outcomes; and (3) renal outcomes. Our review was guided by the following research question: In a population with T2D, treated with SGLT2i or GLP1-RA, what are the biomarkers associated with heterogenous treatment effects in glycaemic, CVD, and renal outcomes? Each of the three outcomes were evaluated separately for each of the two drug classes for a total of 6 sub-studies. Given the heterogeneity of the T2D population, we anticipated that we would find one or more biomarkers modifying the effects of SGLT2i and GLP1-RA.

The Precision Medicine in Diabetes Initiative (PMDI) was established in 2018 by the American Diabetes Association (ADA) in partnership with the European Association for the Study of Diabetes (EASD). The ADA/EASD PMDI includes global thought leaders in precision diabetes medicine who are working to address the burgeoning need for higher quality, individualized diabetes prevention and care through precision medicine[8]. This systematic review is written on behalf of the ADA/EASD PMDI as part of a comprehensive evidence evaluation in support of the 2nd International Consensus Report on Precision Diabetes Medicine[9].

We find that a majority of the papers identified by our review have methodological limitations precluding robust assessment of treatment effect heterogeneity. For SGLT2-inhibitors, multiple observational studies suggest lower renal function as a predictor of lesser glycaemic response, while markers of reduced insulin secretion predict lesser glycaemic response with GLP1-receptor agonists. For both therapies, multiple post-hoc analyses of randomized control trials (including trial meta-analysis) identify minimal clinically relevant treatment effect heterogeneity for cardiovascular and renal outcomes.

## Methods

We conducted a systematic review according to the Preferred Reporting Items for Systematic Reviews and Meta-Analyses (PRISMA) guidelines[10]. The protocol was pre-registered (PROSPERO registration number: CRD42022303236). As above, our review was guided by the following research question: In a population with T2D, treated with SGLT2i and GLP1-RA, what are the biomarkers associated with heterogenous treatment effects in glycaemic, CVD, and renal outcomes?

**Search strategy**. The search strategy for this review was developed for each drug class (SGLT2i and GLP1-RA) and outcome (glycaemic, cardiovascular, and renal) to capture studies specifically evaluating treatment effect heterogeneity associated with demographic, clinical, and biological features in people with type 2 diabetes. Terms for drug class (SGLT2i or GLP1-RA) and individual generic names of licensed drugs within each class (e.g. 'empagliflozin') were included. Potential effect modifiers of interest comprised age, sex, ethnicity, clinical features, routine blood tests, metabolic markers, and genetics; all search terms were based on medical subject sub-headings (MeSH) terms and are reported in Supplementary Note 1. SGLT2i and GLP1-RA were evaluated at drug class level, and we did not aim to identify within-class heterogeneity in treatment effects. Electronic searches were performed in PubMed and Embase by two independent academic librarians in February 2022. Forwards and backwards citation searching was conducted but grey literature and white papers were not searched.

**Inclusion criteria**. To be included, studies were required to meet the following criteria: full-text English-language publications of RCTs, meta-analyses, post-hoc analyses of RCTs, pooled cohort analyses, prospective and retrospective observational analyses published in peer-reviewed journals; adult populations with type 2 diabetes taking at least one of either SGLT2i or GLP1-RA with sample size >100 for the active drug of interest; at least a 4-month potential follow up period (chosen pragmatically as a suitable time length over which changes in glycaemic response could be assessed) after initiation of the drug class of interest; randomized control trials (RCTs) required a comparison against placebo or an active comparator anti-hyperglycaemic drug (observational studies did not require a comparator group); a pre-specified aim of the study must be to examine heterogeneity in treatment outcome, such as biomarker-treatment interactions, stratified analyses, or heterogeneity-focused machine learning approaches; and the study must report differential effects of the drug class on an outcome of interest (see Outcomes section below) with respect to a biomarker. All individual trial or observational cohorts included in a meta-analysis or pooled cohort analysis must have met the inclusion criteria stated above.

We further excluded studies based on the following criteria: studied type 1 or other forms of non-type 2 diabetes; included

children/minors; inpatient studies; conference proceeding abstracts, editorials, opinions papers, book chapters, clinical trial registries, case reports, commentaries, narrative reviews, or non-peer reviewed studies; did not adequately adjust for confounders (individual RCTs and observational studies only, this criteria was not applied for meta-analyses and pooled cohort analyses); did not address the question of treatment response heterogeneity for biomarkers of interest.

Titles and abstracts were independently screened by pairs of research team members to identify potentially eligible studies; these were then independently evaluated for inclusion in the full-text review. Any discrepancies were discussed with a third author until reaching consensus. Discrepancies were discussed as part of larger group meetings to ensure consistency in decisions across reviewer pairs.

**Data extraction and quality assessment**. Pairs of authors independently reviewed the main reports and supplementary materials and extracted the following data for each of the included papers: publication (PMID, journal, publication year, first author, title, study type); study (setting and region, study time period, follow up period); population (overall characteristics, ethnicity); intervention (drug class, specific therapies, treatment/comparator arm sizes); statistical analysis (outcome, outcome measurement, subgroups/predictors analysed with respect to biomarkers, statistical model, covariate set); and results (relevant figures and tables, main findings, methodology, quality). Covidence systematic review software[11] was used for data extraction.

After data were extracted, information was synthesized by drug class and outcome and further examined by biomarkers or subgroups analyzed within each study. Results were extracted within these subsections and summarized for each paper, where general trends in results for each subsection were outlined.

Risk of bias evaluations were conducted alongside the data extraction by each pair of authors, using the Joanna Briggs Institute (JBI) Critical Appraisal Tool for Cohort Studies[12] for all included research papers. This was used to determine the extent of bias within the study's design, execution, and analysis, specifically within the population, outcome measurements, and statistical modelling. The Cohort studies tool was applied for all studies as we did not identify any individual RCTs designed to specifically examine treatment effect heterogeneity, and all included RCT meta-analyses represent post-hoc rather than pre-specified analyses. Further detail on the risk of bias can be seen in Supplementary Figs. 1 and 2. Additionally, the Grading of Recommendations, Assessment, Development, and Evaluations (GRADE) framework[13,14] was applied at the outcome level for each drug class to determine the quality of evidence and certainty of effects for these subsections; an overall GRADE evaluation for all evidence was also provided.

**Outcomes**. Three outcome categories were assessed in the included studies: (1) changes in HbA1c from baseline associated with treatment; (2) CVD outcomes limited to cardiovascular (CV)-related death, non-fatal myocardial infarction, non-fatal stroke, hospitalization for angina, coronary artery bypass graft, percutaneous coronary intervention, hospitalization for heart failure, carotid endarterectomy, and peripheral vascular disease; and (3) renal outcomes including development of chronic kidney disease (including end-stage renal disease, ESRD), and longitudinal changes in markers of renal function including eGFR/creatinine and albuminuria. Specific measurements and procedures for each category of outcome varied across the included studies. Summaries of the included papers assessing each

outcome for each drug class are reported in Supplementary Tables 1-8.

**Reporting summary**. Further information on research design is available in the Nature Portfolio Reporting Summary linked to this article.

## Results

**Literature search and screening results**. Figures 1 and 2 depict the outcomes of the study screening processes for SGLT2i (Fig. 1) and GLP1-RA (Fig. 2).

For SGLT2i, a total of 3415 unique citations underwent title and abstract screening. A total of 3076 were determined to not meet the pre-defined eligibility criteria. The remaining 339 full-text articles were screened, through which process 238 articles were excluded. The most common reasons for exclusion were: studies did not report on the heterogeneity of treatment response (126 studies), studies reported only univariate or unadjusted associations (41 studies), and studies did not meet inclusion criteria (64 studies). In total, 101 studies were identified for inclusion based on the systematic search.

For GLP1-RA, a total of 2270 unique citations underwent title and abstract screening. 2109 were determined to not meet the pre-defined eligibility criteria. The remaining 161 full-text articles were screened, through which process 86 articles were excluded. The most common reasons for exclusion were: studies did not meet inclusion criteria (39 studies), studies reported only univariate or unadjusted associations (26 studies), and studies did not report on the heterogeneity of treatment response (17 studies). In total, 75 studies were identified for inclusion.

**Description of included studies**. Included studies for CVD and renal outcomes were predominantly secondary analyses of industry-funded placebo-controlled trials (RCT), or meta-analyses of these trials, with a smaller number of observational studies. For glycaemic outcomes, most studies were observational. Supplementary Tables 1-8 show all included studies for GLP1-RA and SGLT2i, split by glycaemic, CVD, and renal outcomes, and

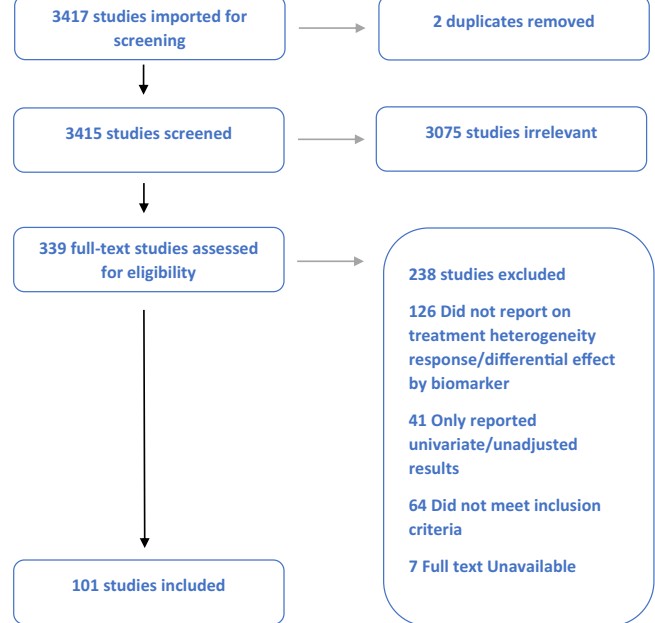

**Fig. 1 Study screening and attrition flow diagram (PRISMA) for SGLT2-inhibitor studies.** Study screening and attrition flow diagram (PRISMA) for SGLT2-inhibitor studies.

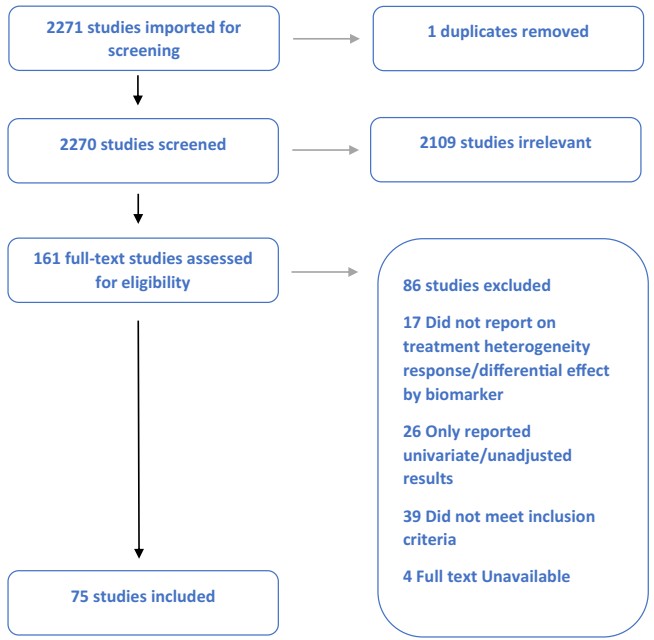

**Fig. 2 Study screening and attrition flow diagram (PRISMA) for GLP1-receptor agonist studies.** Study screening and attrition flow diagram (PRISMA) for GLP1-receptor agonist studies.

including information on study population size, examined biomarkers, and notable findings. Summaries of the major individual RCTs that were included in meta-analyses are detailed in Supplementary Tables 9 and 10.

**SGLT2i, GLP1-RA, and glycaemic outcomes.** Study quality for assessment of heterogenous treatment effects of both drug classes was variable with strong methodological limitations for the study of predictors of glycaemia treatment response common. A core weakness with many studies was a lack of head-to-head comparisons between therapies, which is required to separate broader prognostic factors (that predict response to any glucose-lowering therapy) from drug-specific factors that are associated with differential treatment response. Put otherwise, even when data suggested that a biomarker was associated with glycaemic response, it was not clear if this factor was helpful for choosing between therapies due to the lack of an active comparator.

Other common methodological weaknesses included the use of arbitrary subgroups (rather than the assessment of continuous predictors), small numbers in comparator subgroups that limited statistical power, dichotomized outcomes (responder analysis), multiple testing, and lack of adjustment for key potential confounders.

*SGLT2i.* Of 27 studies that met our inclusion/exclusion criteria, 9 observational studies (usually retrospective analysis of healthcare records), 5 post-hoc analysis of individual RCTs, 10 pooled analyses of individual data from multiple RCTs, and 3 RCT meta-analyses were included (Supplementary Table 7). All included studies assessed routine clinical characteristics and routinely measured clinical biomarkers (Table 1). No pharmacogenetic, or, with the exception of one study of HOMA-B[15], non-routine biomarker studies were identified.

A key finding across multiple studies including appropriately adjusted analysis of RCT and observational data was that HbA1c reduction with SGLT2i is substantially reduced with lower eGFR[16–22]. In pooled RCT data for canagliflozin 300 mg, 6-month HbA1c reduction was estimated to be 11.0 mmol/mol

for participants with eGFR ≥90 mL/min/1.73 m², compared to 6.7 mmol/mol for those with eGFR 45-60[22]. With empagliflozin 25 mg, 6-month HbA1c reduction was 9.6 mmol/mol at eGFR ≥90, and 4.3 mmol/mol at eGFR 30-60[19].

A further finding confirmed by multiple robust studies is that in keeping with other glucose-lowering agents, higher baseline HbA1c is associated with greater HbA1c lowering with SGLT2 inhibitors, including verses placebo[15,21,23–27]. Active comparator studies suggested that higher baseline HbA1c may predict greater relative HbA1c response to SGLT2i therapy in comparison to DPP4i and sulfonylurea therapy[15,25,26]. Notably, an individual participant data meta-analysis of two RCTs showed greater improvement with empagliflozin (6-month HbA1c decline per unit higher baseline HbA1c [HbA1c slope] −0.49% [95%CI −0.62, −0.37] compared to sitagliptin (6-month HbA1c slope −0.29% [95%CI −0.42, −0.15]) and glimepiride (12-month HbA1c slope: empagliflozin -0.52% [95%CI −0.59, −0.44]; glimepiride −0.32% [95%CI −0.39, −0.25])[25].

A number of studies assessing differences in glycaemic response to SGLT2i by ethnicity suggest that initial glycaemic response to this medication class does not vary by ethnicity[28–32]. Similarly, many studies also showed that response did not vary meaningfully by sex. Some studies suggested older age may be associated with reduced glycaemia response; however, analyses usually did not adjust for eGFR which may confound this association, as eGFR declines with age[17,23,32–35].

*GLP1-RA.* Of 49 studies that met our inclusion/exclusion criteria, 24 observational studies, 6 post-hoc analysis of individual RCTs, and 19 meta-analyses were included (Supplementary Table 8). The majority of included studies assessed routine clinical characteristics and routinely measured clinical biomarkers, although 3 studies evaluated genetic variants, and 15 studies evaluated non-routine biomarkers (Table 1).

Studies consistently identified baseline HbA1c as a predictor of greater HbA1c response. For other clinical features, the strongest evidence was that, in many observational studies, markers of lower insulin secretion (including longer diabetes duration [or proxies such as insulin treatment], lower fasting C-peptide, lower urine C-peptide-to-creatinine ratio, and positive glutamic acid decarboxylase (GAD) or islet antigen 2 (IA2) islet autoantibodies) were associated with lesser glycaemic response to GLP1-RA[36–49]. One large prospective study (n=620) observed clinically relevant reductions in HbA1c response with GLP1-RA in individuals with GAD or IA2 autoantibodies (mean HbA1c reduction −5.2 vs. −15.2 mmol/mol without autoantibodies) or C-peptide <0.25 nmol/L (mean HbA1c reduction −2.1 vs. −15.3 mmol/mol with C-peptide >0.25 nmol/L). In contrast, post-hoc RCT analyses have found T2D duration[50] and beta-cell function[51,52] do not modify glycaemic outcomes. This may reflect trial inclusion criteria as included participants had relatively higher beta-cell function, and were less-commonly insulin-treated, compared with the observational cohorts[51].

Few studies contrasted HbA1c outcome for GLP1-RA versus a comparator drug. One meta-analysis showed a greater HbA1c reduction with the GLP1-RA liraglutide compared to other antidiabetic drugs (sitagliptin, glimepiride, rosiglitazone, exenatide, and insulin glargine) across all baseline HbA1c categories (n = 1804)[53], a finding supported for the GLP1-RA dulaglutide compared to glimepiride and insulin glargine[54].

Overall, there was no consistent evidence for effect modification by body mass index (BMI), sex, age or kidney function, with studies reporting contrasting, or null, associations for these clinical features[39,40,44–46,50,54–64]. In comparative analysis, one large observational study found that markers of insulin resistance (including higher HOMA-IR, BMI, fasting triglycerides, and

**Table 1 Summary of evidence for treatment effect heterogeneity for SGLT2-inhibitor and GLP1-receptor agonist therapies for glycaemic outcomes.**

| Glycaemic outcomes / Biomarker | GLP1-RA — GRADE EVIDENCE C | | | SGLT2i — GRADE EVIDENCE B | | | Plain language summary |
|---|---|---|---|---|---|---|---|
| | N (observational) | N (RCT) | N (Meta-analysis and pooled RCT) | N (observational) | N (RCT) | N (Meta-analysis and pooled RCT) | |
| Glycaemia Biomarkers | 18 [37-41,45,47,48,55,61,62,140-145] | 11 [32,44,46,50,56,59,63,146-149] | 1 [53] | 4 [24,150-152] | 2 [27,32] | 3 [15,25,26] | Greater glycaemic response for both SGLT2i and GLP1-RA is seen in individuals with higher baseline HbA1c, No studies are available to comparing the relative efficacy of SGLT2i to GLP1-RA at different baseline HbA1c levels. |
| Renal Function | 6 [39,40,45,54,62,142] | 4 [32,44,59,63] | 1 [64] | 2 [151,152] | 3 [16,18,32] | 4 [19,20,22,26] | GLP1-RA: no evidence that renal function alters glycaemic response. SGLT2i: Lesser glycaemic response for renally impaired patients and those with lower baseline kidney function (eGFR). |
| BMI | 15 [37-41,45,48,54,57,58,61,62,140,142,143] | 11 [32,44,46,56,59,60,63,147,148,153,154] | 0 | 4 [21,24,150,151] | 2 [27,32] | 2 [22,26] | No consistent evidence of significant modifying effects of BMI on glycaemic response for either SGLT2i or GLP1-RA. |
| Age | 13 [37-41,45,47,54,61,62,140,142,143] | 7 [32,44,46,59,63,148,155] | 0 | 4 [17,24,150,151] | 1 [32] | 6 [22,26,33-35,156] | GLP1-RA: No evidence that age alters glycaemic response with GLP1-RA. SGLT2i: some studies suggest older age may be associated with reduced glycaemia response, however, analyses usually did not adjust for eGFR which may confound this association as eGFR declines with age. |
| Diabetes duration | 15 [37-42,45,47,48,54,61,62,140,142,144] | 5 [32,46,50,59,63] | 1 [49] | 3 [150-152] | 1 [32] | 1 [26] | SGLT2i: No consistent effect of diabetes duration on glycaemic response. GLP1-RA: Longer diabetes duration (or proxies such as insulin treatment) associated with lesser glycaemic response. |
| Sex | 13 [37-41,45,48,54,55,62,140,142,143] | 6 [32,46,50,56,59,63] | 0 | 4 [24,30,150,151] | 1 [32] | 3 [22,26,156] | No consistent evidence of significant modifying effects of sex on glycaemic response for either SGLT2i or GLP1-RA. |
| Ethnicity | 2 [40,67] | 5 [32,59,63,65,68] | 1 [66] | 1 [30] | 1 [32] | 4 [15,26,28,29] | No consistent evidence of differences in glycaemic response across ethnic groups for either SGLT2i or GLP1-RA. |
| Genetics | 2 [43,140] | 1 [69] | 0 | 0 | 0 | 0 | SGLT2i: no studies examined genetic factors. GLP1-RA: Two small studies suggest variants rs163184 and rs10305420 may be associated with lesser response in individuals of Chinese ethnicity. |
| Non-routine biomarkers | 12 [37,38,40,41,45,48,54,57,62,142,143,157] | 3 [44,63,158] | 0 | 0 | 1 [159] | 1 [15] | SGLT2i: No evidence of heterogeneity in treatment response for measures of insulin secretion and insulin resistance, or for patients with obstructive sleep apnea. GLP1-RA: Observational studies suggest markers of lower insulin secretion including lower fasting C-peptide, lower urine C-peptide-to-creatinine ratio, and positive GAD or IA2 islet autoantibodies are associated with lesser glycaemic response. In contrast, post-hoc RCT analyses found insulin secretion does not modify glycaemic outcome. This may reflect trial inclusion criteria as participants had relatively higher beta-cell function compared with the observational cohorts. |

HDL) do not alter GLP1-RA response, but are associated with lesser DPP4-inhibitor response[57].

There was limited evidence for differences by ethnicity. One large pooled RCT analysis ($N = 2355$) suggested greater HbA1c response in Asian participants compared to those of other ethnicities, but other studies have not identified differences in response across ethnic groups[65–68]. Similarly, limited studies evaluated pharmacogenetics, although two small studies suggest variants rs163184 and rs10305420, but not rs3765467, may be associated with lesser response in Chinese patients[43,69].

### SGLT2i, GLP1-RA and cardiovascular outcomes

*SGLT2i: Evidence from clinical trials.* Of 65 studies, 58 were post-hoc meta-analysis of RCTs or meta-analysis of multiple RCTs. Heart failure was common as a secondary outcome. The majority of studies were derived from EMPA-REG[70] and the CANVAS program[71], although more recent meta-analyses included up to 12 cardiovascular outcome trials (CVOTs) with different inclusion criteria, treatments, primary outcomes, and follow-up duration (Supplementary Table 9). Most studies included only participants with established CVD or elevated cardiovascular risk, although some studies were restricted to patients with pre-existing heart failure or chronic kidney disease. While most CVOTs and meta-analyses included only patients with type 2 diabetes, some meta-analyses also included data from patients without diabetes in the EMPEROR-P[72], EMPEROR-R[73], DAPA-HF[74] and DAPA-CKD[75] RCTs. Studies primarily focused on relative rather than absolute treatment effects and one of two primary outcomes: 3-point MACE which was a composite of cardiovascular death, non-fatal MI, and non-fatal stroke; or composite heart failure outcomes including hospitalized heart failure and cardiovascular death. The longest duration of follow-up was in the CANVAS CVOT with a median follow-up of 5.7 years, while most other included CVOTs had durations of 1 to 4 years.

On average, in relative terms, SGLT2i reduce the risk of cardiovascular disease (MACE) by 10% (HR 0.90 [95%CI 0.85, 0.95]), and heart failure hospitalization by 32% (HR 0.68 [95%CI 0.61, 0.76]) in individuals with or at high-risk of CVD[2]. The majority of meta-analyses of CVOTs found no significant interactions for MACE or heart failure outcomes across a variety of biomarkers (Table 2; Supplementary Table 1). Several meta-analyses found no interactions by age, sex, and adiposity for MACE or heart failure outcomes. Four meta-analyses examined interactions by race for MACE outcomes and found no interactions. Three meta-analyses consistently identified a greater relative heart failure benefit of SGLT2i in people of Black and Asian ethnicity[76–78] (HR SGLT2i versus placebo 0.60 [95% CI 0.47, 0.74]) compared to White individuals (HR 0.82 [95% CI 0.73, 0.92])[76], however, one meta-analysis reported no difference between White and non-White individuals[79].

Contemporary meta-analysis incorporating the CREDENCE and VERTIS-CV trials alongside EMPA-REG, CANVAS, and DECLARE suggests history of CVD does not modify the efficacy of SGLT2i for MACE[2,80]. One meta-analysis suggests heart failure severity modifies the efficacy of SGLT2i's for heart failure outcome (composite outcome of cardiovascular death or hospitalization for heart failure) with greater efficacy in patients with NYHA heart failure class II (HR SGLT2i versus placebo 0.66 [95%CI 0.59, 0.74]) than class III or IV (HR 0.86 [95%CI 0.75, 0.99])[77]. Other meta-analyses that examined treatment effect heterogeneity using heart failure history as a binary predictor did not find significant interactions[2,81].

A recent meta-analysis[82] that included 6 CVOTs of patients with diabetes and 4 CVOTs of patients with and without diabetes found that eGFR did not alter the relative benefit of SGLT2 inhibitors for MACE and heart failure outcomes;[2,77,81,83–85] however, a greater relative benefit was reported for MACE in those with higher baseline albuminuria (ACR>300 mg/g HR 0.74 [95%CI 0.66, 0.84]; ACR 30-300 mg/g HR 0.95 [95%CI 0.82, 1.10]) ACR<30 mg/g HR 0.87 [95%CI 0.77, 0.98]).

We identified many secondary analyses of single CVOTs, which largely found no interactions by biomarkers (Supplementary Table 1). Single studies identified potential effect modification for MACE by history of CVD[86], and obesity[87], and history of heart failure for heart failure outcome[88], but these associations were not replicated across the other studies or in multi-RCT meta-analyses. In a secondary analysis of CANVAS, participants with higher levels of biomarkers of cardiovascular stress (high-sensitivity cardiac troponin T (hs-cTnT), soluble suppression of tumorigenesis-2 (sST2), and insulin-like growth factor binding protein 7 (IGFBP7)) had greater relative benefit for MACE; for a multimarker score summing high levels of these 3 biomarkers, the relative benefit of SGLT2i for no abnormal biomarkers was HR: 0.99 [95% CI: 0.66–1.49], 1 abnormal biomarker HR: 1.34 [95% CI: 0.94–1.89], 2 abnormal biomarkers HR: 0.61 [95% CI: 0.45–0.82]), and 3 abnormal biomarkers HR: 0.46 [95% CI:0.18–1.17]; $P_{interaction\ trend} = 0.005$)[89]. Unlike meta-analyses, studies based on single RCTs typically performed multivariable adjustment for potential confounders.

*GLP1-RA: Evidence from clinical trials.* Of the 35 studies that investigated heterogeneity in the effect of GLP1-RAs on cardiovascular health and met our inclusion criteria, 15 were meta-analyses of RCTs or pooled analyses of multiple RCTs, 15 were post-hoc analyses of RCTs, and 5 were observational studies (Supplementary Table 2). Most studies used data collected from the LEADER, SUSTAIN 6, and EXSCEL trials, however in total the data from 7 CVOTs were used (Supplementary Table 10). The majority of these CVOTs investigated the effect of us CVD on the cardiovascular efficacy of GLP1-RAs using 3-point MACE as a primary outcome, and with heart failure being a common secondary outcome, focusing on relative rather than absolute benefit. The population of 6 of the 7 CVOTs had established CVD or high CVD risk. The CVOT with the longest median follow-up was REWIND with a median follow-up of 5.4 years, and the median follow-up of the other CVOTs ranged from 1 to 4 years.

Contemporary meta-analysis data suggests GLP1-RA reduces the relative risk of cardiovascular disease (MACE) by 14% (HR 0.86 [95%CI 0.80-0.93]), and heart failure hospitalization by 11% (HR 0.89 [95%CI 0.82, 0.98]) compared to placebo[3]. Several large meta-analyses examining heterogenous treatment effects in placebo-controlled CVOTs have been conducted for GLP1-RA[76,83,84,90–97], with the majority of studies focusing on whether prior established CVD modifies the relative effect of GLP1-RA on MACE and/or heart failure. Two meta-analyses reported the relative MACE benefit of GLP-RA may be restricted to those with established CVD[83,90], the largest of which included 7 RCTs and reported a 14% relative risk reduction with GLP1-RA specific to individuals with established CVD (with CVD: HR 0.86 [95%CI 0.80, 0.93]; at high-risk of CVD: HR 0.94 [95% CI 0.82, 1.07])[83]. However, this risk difference is not conclusive and has not been replicated in other meta-analyses and pooled RCT analyses[91–93,98,99], including an individual participant level re-analysis of the SUSTAIN and PIONEER RCTs which evaluated baseline CVD risk as a continuous rather than subgroup-level variable[100].

Differential relative effects of GLP1-RAs on MACE have been reported by ethnicity in two out of three meta-analyses:[76,83,90] one showed a benefit of GLP1-RA treatment compared to placebo in Asian (HR 0.76 [95%CI 0.61, 0.96]) and Black (HR 0.77 [95%

**Table 2 Summary of evidence for treatment effect heterogeneity for SGLT2-inhibitor and GLP1-receptor agonist therapies for cardiovascular outcomes (including heart failure).**

| Cardiovascular disease (CVD) Biomarker | GLP1-RA GRADE EVIDENCE B | | | SGLT2i GRADE EVIDENCE B | | | Plain language summary |
|---|---|---|---|---|---|---|---|
| | N (observational) | N (RCT) | N (Meta-analysis and pooled RCT) | N (observational) | N (RCT) | N (Meta-analysis and pooled RCT) | |
| Race/ethnicity | 0 | 0 | 3[76,83,90] | 0 | 2[160,161] | 4[76,78,83,110] | No heterogeneity by race/ethnicity for SGLT2is; Potential increased cardiovascular benefit in Asians associated with GLP1-RA use, but results are inconsistent. |
| History of CVD | 3[105-107] | 8[125,162-167] | 7[83,90-95,98,99] | 4[104-106,109] | 3[86,168,169] | 5[2,80,83,95,112] | No consistent impact on SGLT2i or GLP1-RA outcomes. |
| Age | 2[107,108] | 3[121,165,170] | 3[83,92,94] | 2[104,109] | 1[171] | 4[78-80,83] | No consistent heterogeneity by age for SGLT2is; No evidence of age effect for GLP1-RAs. |
| Sex | 3[102,107,108] | 1[165] | 5[83,90,92,94,96,99] | 4[102,104,109,110] | 3[171-173] | 4[79,83,174,175] | No consistent heterogeneity by sex for SGLT2is or GLP1-RAs. |
| Renal function | 1[108] | 2[127,176] | 5[83,84,90,92,94] | 2[104,109] | 6[16,117-119,171,177] | 5[2,82-84,95] | No consistent heterogeneity by renal function on cardiovascular outcomes for SGLT2is; No evidence of heterogeneity by renal function on cardiovascular outcomes for GLP1-RAs. |
| BMI | 0 | 1[126] | 5[83,90,92,94,97] | 1[110] | 2[87,171] | 3[79,83,97] | No consistent heterogeneity by BMI for SGLT2is; Some inconsistent evidence suggests that higher baseline BMI may improve cardiovascular efficacy of GLP1-RAs. |
| Genetics | 0 | 0 | 0 | 0 | 0 | 0 | The greater benefit of SGLT2i in those with high levels of 3 biomarkers: hs Cardiac Troponin T, soluble suppression of tumorigenesis-2 (sST2), and insulin-like growth factor binding protein 7 (IGFBP7) levels |
| Non-routine biomarkers | 0 | 0 | 0 | 0 | 5[89,178-181] | 0 | |
| **Heart Failure (HF)** GRADE EVIDENCE B | | | | GRADE EVIDENCE B | | | |
| Ethnicity | 0 | 1[82] | 1[76] | 0 | 4[160,161,183,184] | 4[76-79] | Possibly greater relative benefit of SGLT2i in Asian and Black compared to white ethnicity; Potential increased efficacy of GLP1-RAs in Asian ethnicity. |
| Age | 0 | 2[165,170] | 1[94] | 1[104] | 1[183] | 3[79,81] | No heterogeneity by age for SGLT2is or GLP1-RAs. |
| Sex | 0 | 2[182] | 1[94] | 2[102,104] | 3[172,173,183] | 3[79,81,175] | No heterogeneity by sex for SGLT2is or GLP1-RAs. |
| BMI | 0 | 0 | 1[94] | 0 | 3[87,183,184] | 2[79,81] | No consistent heterogeneity by BMI for SGLT2is or GLP1-RAs. |
| History of CVD | 2[105,106] | 3[165,166,182] | 3[93-95] | 4[101,104-106] | 4[86,168,169,185] | 4[2,77,81,85] | No consistent heterogeneity by CVD history for SGLT2is or GLP1-RAs. |
| History of HF | 0 | 0 | 0 | 1[106] | 1[88] | 4[2,77,81,85] | No consistent heterogeneity by HF history for SGLT2is; No analysis on heterogeneity by HF history was performed for GLP1-RAs. |
| HF severity/score | 0 | 0 | 0 | 1[103] | 3[163,186,187] | 1[77] | Greater relative benefit of SGLT2i in those with NYHA class II vs class III/IV in one meta-analysis; No analysis of heterogeneity by HF severity/score performed for GLP1-RAs. |
| Renal function | 0 | 0 | 1[94] | 2[103,104] | 6[16,117-119,177,183] | 5[2,77,81,82,85] | No consistent heterogeneity in renal function. A single meta-analysis showed greater SGLT-2 benefit with lower eGFR and higher ACR; No evidence for heterogeneity by renal function for GLP1-RAs. |
| Genetics | 0 | 0 | 0 | 0 | 0 | 0 | No heterogeneity across a variety of non-routine biomarkers. No analysis of heterogeneity by non-routine biomarkers was performed for GLP1-RAs. |
| Non-routine biomarkers | 0 | 0 | 0 | 0 | 6[89,159,178-180,188] | 0 | |

CI 0.59, 0.99]) individuals, but not in White individuals (HR 0.95 [95%CI 0.88, 1.02]);[90] the second showed a significantly greater benefit of GLP1-RA for MACE in Asian compared to White individuals (HR Asian 0.68 [95%CI 0.53, 0.84]; White 0.87 [95% 0.81, 0.94])[76]. For other clinical features including sex, BMI/obesity, baseline kidney disease, duration of diabetes, baseline HbA1c, background glucose lowering medications, and prior history of microvascular disease, the overall body of evidence from meta-analyses does not provide robust evidence to support differential effects of GLP1-RA on CVD outcomes (Table 2).

*SGLT2i and GLP1-RA: Evidence from observational studies.* 10 observational studies met our inclusion criteria, with studies primarily reporting relative rather than absolute risk differences[101–110]. These studies comparing SGLT2i and GLP1-RA individually with other oral therapies (predominantly DPP4i) generally reported average relative benefits for CVD and heart failure outcomes in-line with placebo-controlled trials, with no consistent pattern of subgroup level differences across studies (Supplementary Tables 1 and 2).

A few observational studies compared SGLT2i and GLP1-RA CVD outcomes. In a US claims-based study with follow-up to two years (n = 47,343), Htoo et al. [106] reported a higher relative risk of MACE with SGLT2i compared to GLP1-RA specific to individuals without CVD and heart failure (Relative risk [RR] 1.31 [95% CI 1.09, 1.56]), and a higher risk of stroke with SGLT2i versus GLP1-RA specific to individuals without CVD (No CVD without heart failure: RR 1.62 [95%CI 1.10, 2.38]; No CVD with heart failure: RR 3.30 [95%CI 1.22, 8.97]). In contrast, over a median follow-up of 7 months, Patorno et al. [105] reported a lower relative risk of myocardial infarction with SGLT2i compared to GLP1-RA in US claims data specific to individuals with a history of CVD (n=156,825; HR 0.83 [95%CI 0.74, 0.93] with history of CVD; HR 1.13 [95%CI 1.00, 1.28] without history of CVD), with no differences in stroke outcomes irrespective of CVD status. Both studies reported a consistent benefit of SGLT2i over GLP1-RA for heart failure. Raparelli et al. [102] identified potential differences by sex in the Truven Health MarketScan database (n=167,341): compared to sulfonylureas and over a median follow-up of 4.5 years, there was a greater relative reduction with GLP1-RA for females (HR 0.57 [95%CI 0.48, 0.68]) compared to males (HR 0.82 [95%CI 0.71, 0.95]), but a similar benefit for both sexes with SGLT2i (females HR 0.58 [95%CI 0.57, 0.83]; males HR 0.69 [95%CI 0.57, 0.83]).

## SGLT2i, GLP1-RA, and renal outcomes
*SGLT2i: Evidence from clinical trials.* A total of 29 studies met our inclusion criteria. These included 20 post-hoc analyses of individual RCTs, 7 trial meta-analyses (Supplementary Table 4), and 2 analyses of observational data. All of the post-hoc RCT analyses and all but 1 of the meta-analyses used only data from the 12 SGLT2i cardiovascular/renal RCTs shown in Supplementary Table 9, which therefore provided most of the evidence in our review. These trials included people with type 2 diabetes with and without pre-existing cardiovascular disease, and had composite renal endpoints incorporating two or more of the following (which differed between trials): changes in eGFR/serum creatinine, end-stage renal disease, changes in urine albumin:creatinine ratio (ACR), and/or death from renal causes. Most studies assessed routine clinical characteristics, especially renal function as measured by eGFR or urine ACR or a combination of both. In addition, 4 post-hoc RCT analyses examined non-routine plasma biomarkers. We found no genetic studies (Table 3).

On average, SGLT2i have a relative benefit for a number of renal outcomes including kidney disease progression (HR 0.63, 95%CI 0.58,0.69) and acute kidney injury (HR 0.77, 95%CI 0.70,

0.84)[4]. Placebo-controlled trial meta-analyses of subgroups found no evidence for heterogeneity of SGLT2i treatment effects on relative renal outcomes by age[79], use of other glucose-lowering drugs[79], use of blood pressure/cardiovascular medications[79,111], blood pressure[79], BMI[79], diabetes duration[79], White race[79], history of cardiovascular disease or heart failure[2,80] or sex[79].

For baseline eGFR, an early meta-analysis that included EMPA-REG, CANVAS, and DECLARE reported greater effect of SGLT2i on renal outcomes in those with higher eGFR[112] but both a later meta-analysis that added CREDENCE[111] and a recent meta-analysis that added two further studies (SCORED and DAPA-CKD, including some participants without diabetes)[82] showed no effect of baseline eGFR on renal outcomes with SGLT2i. For urine ACR, meta-analyses of subgroups found no evidence for greater SGLT2i effect with higher UACR[2,82,111,113]. Single RCTs found no heterogeneity of treatment effect by eGFR and UACR, or subgroups defined by the combination of these two[114–118], with the exception of Neuen et al. [119] which showed a greater SGLT2i effect in preventing eGFR decline relative to placebo for those with higher UACR, and heterogeneity in a composite renal outcome by UACR. Overall, there was limited or no evidence to support modifying effects of baseline eGFR or UACR on the effect of SGTL2i on renal function outcomes.

A few post-hoc analyses of the CANVAS RCT considered non-routine biomarkers, with most showing no interaction with SGLT2i treatment and renal outcomes. Two RCTs studied the effect of SGLT2i on renal outcomes at differing plasma IGFBP7 levels. One study reported an interaction of IGFBP7 with SGLT2i treatment for progression of albuminuria (>96.5 ng/ml HR 0.64; <=96.5 ng/ml HR 0.95, $P_{interaction}$ = 0.003)[120] but no effect was seen for the composite renal endpoint in two studies[89,120]. The biomarker panel (sST2, IGFBP7, hs-cTnT) that showed a strong interaction with SGLT2i for MACE outcomes (see above) did not show any interaction for renal outcomes[89].

*GLP1-RA: Evidence from clinical trials.* 7 studies met our inclusion criteria: all post-hoc RCT analyses, 6 of individual trials (or multiple trials analysed separately) and 1 pooled analysis of two RCTS (Supplementary Table 5). These studies used data from 5 of the 7 GLP1-RA cardiovascular outcome trials shown in Supplementary Table 10, with renal outcomes only a secondary endpoint. Most of these trials had composite renal endpoints as per the SGLT2i cardiovascular/renal trials, while some examined changes in either eGFR or urine ACR only. All studies assessed routine clinical characteristics, especially renal function as measured by eGFR or urine ACR. No studies of genetics or non-routine biomarkers were identified (Table 3). The overall sample sizes were small and subgroup analyses underpowered to show a subgroup by treatment interaction for renal outcomes.

Overall, GLP1-RA reduce the relative risk of albuminuria over 2 years by 24% versus placebo (HR 0.76 [95% CI 0.73-0.80; P < 0.001), and similarly reduce the relative risk of a 40% reduction in eGFR (HR, 0.86 [95% CI 0.75-0.99]; P = 0.039)[5]. Studies found no heterogeneity of GLP1-RA relative treatment effect by age[121], blood pressure[122,123], diabetes duration[124], history of cardiovascular disease/heart failure[122,125] or use of RAS inhibitors[122]. For BMI, a post-hoc analysis of EXSCEL (Exenatide) found a greater GLP1-RA effect on reducing rate of eGFR decline in those with lower BMI (BMI ≤ 30 kg/m² treatment difference 0.26 mL/min/1.73m²/year [95% CI 0.04, 0.48] vs BMI > 30 kg/m² −0.12 [-0.26, 0.03], $P_{interaction}$ = 0.005)[122]. However, Verma et al.[126] found no significant interaction by BMI subgroup with GLP1-RA treatment for a composite renal outcome in LEADER (Liraglutide) or SUSTAIN 6 (Semaglutide).

For baseline eGFR, a pooled analysis of LEADER and SUSTAIN-6 reported a significant interaction, with lower eGFR

**Table 3 Summary of evidence for treatment effect heterogeneity for SGLT2-inhibitor and GLP1-receptor agonist therapies for renal outcomes.**

| Biomarker | GLP1-RA | | | SGLT2i | | | Plain language summary |
|---|---|---|---|---|---|---|---|
| **Renal (eGFR changes/CKD progression/composite outcomes of these with or without ACR changes)** | **GRADE EVIDENCE B** | | | **GRADE EVIDENCE B** | | | |
| | N (observational) | N (RCT) | N (Meta-analysis and pooled RCT) | N (observational) | N (RCT) | N (Meta-analysis and pooled RCT) | |
| Baseline HbA1c | 0 | 0 | 0 | 0 | 0 | 0 | Generally no relationship between either eGFR or ACR and GLP1-RA benefit. Greater relative benefit of SGLT2i in those with higher eGFR (although inconsistent results with some studies showing no impact and 1 observational study finding the opposite relationship). Generally no relationship between ACR/proteinuria and SGLT2i benefit. |
| Renal Function | 0 | 3[122,127,128] | 1[5] | 2[129,130] | 6[114,115,117–119,127] | 5[2,82,111–113] | |
| BMI | 0 | 2[122,126] | 0 | 1[129] | 2[87,184] | 1[79] | Greater GLP1-RA benefit with lower BMI but not seen consistently. Generally no effect on SGLT2i benefit |
| Age | 0 | 1[121] | 0 | 2[129,130] | 0 | 1[79] | No effect on GLP1-RA or SGLT2i benefit |
| Diabetes duration | 0 | 1[124] | 0 | 0 | 0 | 1[79] | No effect on GLP1-RA or SGLT2i benefit |
| Sex | 0 | 0 | 0 | 1[129] | 1[173] | 1[79] | No effect on SGLT2i benefit |
| Ethnicity | 0 | 0 | 0 | 0 | 2[184,189] | 1[79] | No effect on SGLT2i benefit |
| Genetics | 0 | 0 | 0 | 0 | 0 | 0 | |
| Non-routine biomarkers | 0 | 0 | 0 | 0 | 5[89,120,178–180] | 0 | No effect on SGLT2i benefit |
| Blood pressure/hypertension | 0 | 2[122,123] | 0 | 1[129] | 0 | 1[79] | No effect on GLP1-RA or SGLT2i benefit |
| History of CVD/HF | 0 | 2[122,125] | 0 | 1[129] | 3[168,169,190] | 3[2,80,112] | No effect on GLP1-RA or SGLT2i benefit |
| **Renal (albuminuria changes)** | **GRADE EVIDENCE B** | | | **GRADE EVIDENCE B** | | | |
| Baseline HbA1c | 0 | 0 | 0 | 0 | 0 | 0 | |
| Renal Function | 0 | 2[122,128] | 1[5] | 0 | 0 | 1[113] | Greater GLP1-RA benefit with higher ACR although not seen consistently. No relationship between eGFR and GLP1-RA. |
| BMI | 0 | 1[122] | 0 | 0 | 0 | 0 | No effect on SGLT2i benefit |
| Age | 0 | 0 | 0 | 0 | 0 | 0 | No effect on GLP1-RA benefit |
| Diabetes duration | 0 | 0 | 0 | 0 | 0 | 0 | |
| Sex | 0 | 0 | 0 | 0 | 0 | 0 | |
| Ethnicity | 0 | 0 | 0 | 0 | 1[189] | 0 | No effect on SGLT2i benefit |
| Genetics | 0 | 0 | 0 | 0 | 0 | 0 | |
| Non-routine biomarkers | 0 | 0 | 0 | 0 | 1[120] | 0 | Single trial found greater SGLT2i benefit at higher IGFBP7 |
| Blood pressure/hypertension | 0 | 1[122] | 0 | 0 | 0 | 0 | No effect on GLP1-RA benefit |
| History of CVD/HF | 0 | 1[122] | 0 | 0 | 0 | 0 | No effect on GLP1-RA benefit |

associated with greater GLP1-RA effect in reducing eGFR decline: Semaglutide 1.0 mg vs placebo, eGFR < 60 difference in decline 1.62 ml/min/1.73m²/year vs eGFR > = 60 difference in decline 0.64 ml/min/1.73 m²/year, $P_{interaction} = 0.057$; Liraglutide 1.8 mg vs placebo, eGFR < 60 difference in decline 0.67 ml/min/1.73m²/year vs 0.15 ml/min/1.73 m²/year, $P_{interaction} = 0.008$[5]. However, a study of Exenatide LAR found no treatment heterogeneity for this same outcome by eGFR category[122], and in a further analysis of LEADER, the renal composite endpoint was used with no interaction reported by baseline eGFR category[127]. The overall evidence does not support an effect of baseline eGFR on the relative renal benefit for GLP1-RA as an overall drug class.

For baseline UACR, a pooled analysis of LEADER and SUSTAIN-6[5] and EXSCEL[122] showed a greater benefit of GLP1-RA on eGFR reduction or eGFR slope with higher UACR; however, there was either no significant interaction[5] or no formal interaction test was reported[122]. For ELIXA, Muskiet et al. [128] did not find a significant interaction of UACR category on eGFR decline. A further study found no association between UACR and GLP1-RA effect on reducing a composite renal outcome[127].

Two studies found that GLP1-RAs more effectively reduced UACR in those with higher UACR. In a pooled analysis of LEADER and SUSTAIN-6, those with normal albuminuria had a 20% (95%CI 15%, 25%) reduction in UACR compared to placebo; those with microalbuminuria had a 31% (95%CI 25–37%) reduction; those with macroalbuminuria had a 19% (95%CI 7–30%); $P_{interaction} = 0.021$[5]. In ELIXA, least-squares mean percentage change in UACR was −1.69% (SE 5.10; 95% CI −11.69 to 8.30; $p = 0.7398$) in participants with normoalbuminuria, −21.10% (10.79; −42.25 to 0.04; $p = 0.0502$) in participants with microalbuminuria, and −39.18% (14.97; −68.53 to −9.84; $p = 0.0070$) in participants with macroalbuminuria in favour of lixisenatide; a formal test for interaction was not reported[128]. A third study found no treatment heterogeneity for this same outcome[122].

In summary, the included studies showed conflicting results for renal outcomes of GLP1-RA, though the majority were underpowered to detect heterogenous treatment effects. The most consistent finding was that a higher UACR is associated with greater GLP1-RA reduction in UACR relative to placebo, but this does not translate to benefits in eGFR-defined measures of renal function. There were no other biomarkers that robustly predicted benefit from GLP1-RA for the renal outcomes examined.

*SGLT2i and GLP1-RA: Evidence from observational studies.* There were no observational studies for GLP1-RA and renal outcomes included, and no comparison studies between people treated with GLP1-RA and SGLT2i. Observational studies comparing SGLT2i to other glucose-lowering drugs confirmed the lack of treatment effect heterogeneity associated with age[129,130], use of blood pressure/cardiovascular medications[127], blood pressure (Koh 2021), history of cardiovascular disease[129] and sex[129], but one study in a Korean population found greater SGLT2i benefit on progression to end stage renal impairment with higher BMI (BMI < 25 kg/m2, HR 0.80 (95%CI 0.51, 1.25); BMI ≥ 25 kg/m2 HR 0.27 (0.16, 0.44), $P_{interaction} = 0.002$) and with abdominal obesity compared to without[129]. This is not consistent with results from meta-analysis of RCTs.

## Summary of quality assessment

To evaluate risk of bias, we used the JBI critical appraisal tool for cohort studies as the best flexible tool for the range of studies included. Due to our screening criteria, no manuscripts that passed full text screening were excluded due to risk of bias. The checklist results for the 11 points in the appraisal checklist are shown as a heatmap in Supplementary Figure 1 (**SGLT2i**) and 2 (**GLP1-RA**).

Additionally, the Grading of Recommendations, Assessment, Development, and Evaluations (GRADE) framework was applied at the outcome level for each drug class to determine the quality of evidence and certainty of effects (Table 4)[13]. Overall certainty of evidence was rated as moderate for all outcomes except glycaemia with GLP1-RA which was rated low certainty. This reflects that a larger proportion of the studies included for evaluation of GLP1-RA glycaemia outcomes were observational (24/49). By contrast, for SGLT2i glycaemia outcomes there were 18 RCT/meta-analyses and 9 observational studies. For CVD and renal outcomes, observational studies were limited and the majority of evidence came from industry-funded CVOTs (RCT designs), including post-hoc analyses of individual trials as well as meta-analyses.

## Discussion

This systematic review provides a comprehensive review of observational and RCT-based studies of people with type 2 diabetes, specifically examining heterogenous treatment effects for SGLT2i and GLP1-RA therapies on glycaemic, cardiovascular, and renal outcomes. We assessed evidence for treatment effect modification for a wide range of demographic, clinical and biological features, including pharmacogenetic markers. Each of the three clinical outcomes were evaluated separately for each drug class for a total of 6 sub-studies. Overall, our review identified limited evidence for treatment effect heterogeneity for glycaemia, cardiovascular, and renal outcomes for the two drug classes. We summarize the key findings below.

For glycaemic response, there was high certainty that reduced renal function is associated with lower efficacy of SGLT2i. For GLP1-RA there was moderate certainty that markers of reduced insulin secretion, either directly measured (e.g. c-peptide or HOMA-B) or proxy measures, such as diabetes duration, were associated with reduced glycaemic response to GLP1-RA, although the majority of evidence was from observational studies. As with other glucose-lowering drug classes, a greater glycaemic response with both SGLT2i and GLP1-RA was seen at higher baseline HbA1c. We did not identify any studies examining whether the relative efficacy of SGLT2i compared to GLP1-RA is altered by baseline HbA1c levels. Of note, many of the included studies for HbA1c outcome were observational, meaning findings could potentially reflect biases from differential prescribing behaviour, or regression to the mean, although we did attempt to account for the latter by including adjustment for baseline HbA1c as one of our study inclusion criteria.

For both CVD and heart failure outcomes, RCT meta-analyses do not support differences in the relative efficacy of either GLP1-RA or SGLT2i based on an individuals' prior CVD status. However, this finding should be interpreted cautiously as all RCTs to-date have predominantly included participants with, or at high-risk of, CVD, thereby excluding the majority of the wider T2D population at lower risk. However, meta-analyses suggest (with moderate certainty) that the relative effects of both drug classes may be greater in people of non-White ethnicity. In particular, those of Asian and African ethnicity (compared to Whites) have been shown to have a greater relative benefit for hospitalization for heart failure/CV death (but not MACE) with SGLT2i, and MACE for GLP1-RA.

When evaluating renal outcomes, there was no consistent evidence of treatment heterogeneity for SGLT2i, but for GLP1-RA, there was greater reduction in proteinuria in those with higher baseline proteinuria.

This limited evidence could reflect a true lack of heterogenous treatment effects, but it more likely reflects an absence of clinical

**Table 4 Grading of Recommendations, Assessment, Development, and Evaluations (GRADE) framework summary of findings.**

| Drug class | Outcome | Overall certainty of evidence | Elaboration on evidence certainty | Evidence for specific biomarkers |
|---|---|---|---|---|
| SGLT2i | CVD | Moderate | Majority of evidence from post-hoc analysis of RCTs and RCT-based meta-analysis | - History of prior cardiovascular disease probably does not alter relative benefit (no effect, moderate certainty)<br>- Ethnicity probably does alter relative benefit, with a greater relative heart failure benefit in people of black and Asian ethnicity compared to those of white ethnicity (moderate effect, moderate certainty)<br>- Other biomarkers may not be associated with treatment effect heterogeneity (no effect, low certainty) |
| | Renal | Moderate | Majority of evidence from post-hoc analysis of RCTs and RCT-based meta-analysis | - Biomarkers may not be associated with treatment effect heterogeneity (no effect, low certainty) |
| | Glycaemia | Moderate | Majority of evidence from post-hoc analysis of RCTs and RCT-based meta-analysis | - Lower renal function results in lesser glycaemic response (moderate effect, high certainty)<br>- Other biomarkers may not be associated with treatment effect heterogeneity (no effect, low certainty) |
| GLP1-RA | CVD | Moderate | Majority of evidence from post-hoc analysis of RCTs and RCT-based meta-analysis | - History of prior cardiovascular disease probably does not alter relative benefit (no effect, moderate certainty)<br>- Ethnicity probably does alter relative benefit, with a greater relative CVD benefit in people of black and Asian ethnicity compared to those of white ethnicity (moderate effect, moderate certainty)<br>- Other biomarkers may not be associated with treatment effect heterogeneity (no effect, low certainty) |
| | Renal | Moderate | Majority of evidence from post-hoc analysis of RCTs and RCT-based meta-analysis | - Biomarkers may not be associated with treatment effect heterogeneity (no effect, low certainty) |
| | Glycaemia | Low | Majority of evidence from observational studies | - Lower insulin secretion probably results in lesser glycaemic response (moderate effect, moderate certainty)<br>- Other biomarkers may not be associated with treatment effect heterogeneity (no effect, low certainty) |

studies that were well designed or sufficiently powered to robustly identify and characterise treatment effect heterogeneity. Although five of the six sub-studies we evaluated were evaluated at GRADE B, there were methodological concerns with many of the included studies. As individual RCTs are by design powered only for the main effect of treatment[131], our primary focus when reporting were meta-analyses of post-hoc subgroup analyses of RCTs. However, we found the subgroup analyses in these studies primarily focused on stratification by baseline risk for the outcome in question e.g. baseline HbA1c on glycaemic response, CKD stage or albuminuria on renal outcomes, and CVD risk or established CVD for CVD outcomes. Other common subgroups included those defined by BMI, age, sex or other routinely collected clinical characteristics, with very few studies evaluating non-routine biomarkers or pharmacogenetic markers (as highlighted in Tables 1–3). A major limitation was that studies predominantly focused on conventional approaches to subgroup analysis, with very few studies assessing continuous features (such as BMI) on a continuous scale which is required to maximize power to detect treatment effect heterogeneity[131,132].

It is also important to recognize that almost all the studies evaluating cardiovascular and renal endpoints included in our systematic review focused on the *relative* effect of a biomarker/stratifier on the outcome, as most studies reported a hazard ratio compared with a placebo arm for the outcome of interest (e.g. MACE, incident renal disease). This does not recognize that baseline absolute risk of these endpoints is likely to differ substantially across these strata; so although, for example, there was no difference in relative benefit of an SGLT2i by age, this means that on the absolute scale, benefit will increase with age (as underlying absolute risk increases), and it is this absolute

benefit that should be considered when deciding on whether to initiate SGLT2i treatment.

An important finding of our review is the lack of robust comparative effectiveness studies directly examining treatment effect heterogeneity for these two major drug classes, either head-to-head or compared with other major anti-hyperglycaemic therapies. Insight into effect modification for a single drug class is not sufficient to support the clinical translation of a precision medicine approach. The lack of direct comparisons between therapies obscures the interpretation of biomarkers with regards to whether they function as broad prognostic factors, which may be relevant to any (or at least multiple) drug class, or as markers of heterogenous treatment effects specific to a particular drug class. An evidence base that includes more high-quality studies on heterogeneity in the comparative effectiveness of SGLT2i, GLP1-RA, and other drug classes is needed to advance the field towards clinically useful precision diabetes medicine. For cardiovascular and renal outcomes, these studies need to incorporate both absolute outcome risk and relative estimates of treatment effects in order to usefully inform clinical decision-making. Only when this evidence is available can precision medicine support more individualised treatment decisions, allowing providers to select an optimal therapy from a set of multiple options informed by each medication's risk/benefit profile specific to the characteristics of an individual patient.

We identified the following additional, high-level evidence gaps in our review: (1) Limited head-to-head comparative effectiveness studies examining treatment effect heterogeneity; (2) A lack of robust studies integrating multiple clinical features and biomarkers. The majority of studies only tested single biomarkers one at a time in subgroup analysis; (3) Few studies focused on pharmacogenetics or non-routine biomarkers; (4) Few studies

conducted in low-middle income countries, required for an equitable global approach to precision type 2 diabetes medicine; (5) Few RCT meta-analyses based on individual-level participant data, precluding robust evaluation of between-trial heterogeneity and individual-level confounders; (6) An absence of confirmatory studies. We identified no prospective studies testing a priori hypotheses of potential treatment effect modifiers, or studies conducting independent validation of previously described heterogenous treatment effects; (7) A lack of population-based data representing individuals treated in routine care. As cardiovascular and renal trials have focused on high-risk participants, the benefits of SGLT2i and GLP1-RA for primary prevention is a major unanswered question; (8) Few cardiovascular and renal outcome studies considering treatment effect modification on the absolute as well as relative risk scale; (9) A focus on short-term glycaemic outcomes, with limited studies investigating durability of glycaemic response or time to glycaemic failure.

Of note, several studies published since our data extraction was completed in February 2022 which fill some of the evidence gaps identified in our review, and highlight the clear potential for a precision medicine approach to T2D treatment: the TriMaster study—a precision medicine RCT of SGLT2i, DPP4i and thiazolidinediones (TZD) that established that individuals with higher renal function (eGFR >90 ml/min/1.73 m$^2$) have a greater HbA1c response with SGLT2i vs DPP4i relative to those with eGFR 60–90 ml/min/1.73 m$^2$ [133], a result concordant with our finding that reduced renal function is associated with lower efficacy of SGLT2i; a similarly designed two-way crossover trial in New Zealand which identified a greater relative benefit of TZD therapy compared to DPP4i in people with obesity and/or hypertriglyceridemia;[134] a study using large-scale observational data and post-hoc analysis of individual participant-level data from 14 RCTs that specifically investigated differential treatment effects with SGLT2i and DPP4i, and developed a treatment selection model to predict HbA1c response on the two therapies based on an individuals' routine clinical characteristics;[135] and a robust study across observational and multiple RCTs identifying pharmacogenetic markers of differential glycaemic response to GLP1-RA[136]. In addition, three large trials (AMPLITUDE-O investigating cardiovascular and renal outcomes in 4076 participants with T2D for the GLP-RA efpeglenatide[137], DELIVER investigating worsening heart failure or cardiovascular death in 3131 participants [45% with T2D] for the SGLT2i Dapagliflozin[138], and EMPA-KIDNEY investigating progression of kidney disease or cardiovascular death in 6609 participants [44% with T2D][139]) have recently been published. Although all three are primary RCTs examining average treatment effects rather than treatment effect heterogeneity, and thus would have been ineligible for our review, future meta-analysis studies integrating the results of these and other ongoing SGLT2i and GLP1-RA trials may add to the evidence we have presented.

As our aim was to provide a comprehensive review of these treatments, we did not conduct quantitative analysis of specific biomarkers due to the range of different biomarkers, methodologies, and outcomes evaluated in the included studies. However, this review provides guidance for where future targeted quantitative meta-analysis could be most insightful. In addition, different methods for synthesising the current available evidence, such as conducting an umbrella review, may offer further insights into the current state-of-play of precision Type 2 diabetes treatment.

This review highlights the need for several research priorities to advance our limited understanding of heterogenous treatment effects among individuals with type 2 diabetes. We outline priorities for research to advance the field towards a translational model of evidence-based, empirical precision diabetes medicine

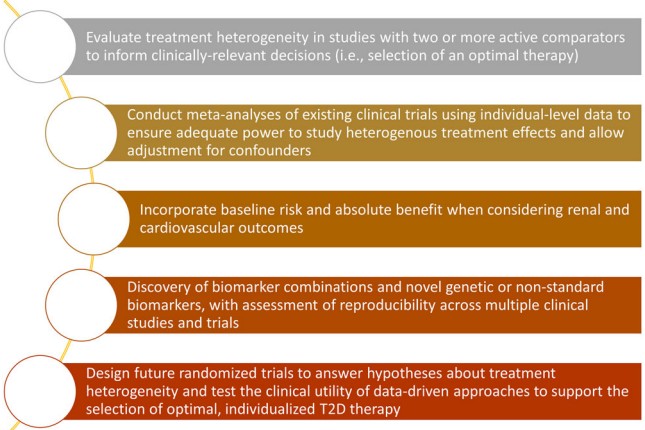

**Fig. 3 Priorities for future research to advance the field towards a translational model of evidence-based, empirical precision diabetes medicine.** Priorities for future research in treatment heterogeneity of diabetes medications as identified by this systematic review.

(Fig. 3), and highlight the recent Predictive Approaches to Treatment effect Heterogeneity (PATH) Statement to guide this research[132]. In the future, with a greater understanding of heterogenous treatment effects and enhanced capacity to predict individual treatment responses, precision treatment in type 2 diabetes may be able to integrate demographic, clinical, biological, or other patient-level features to match individuals to their optimal anti-hyperglycaemic regimen.

## Conclusions

There is limited evidence of treatment effect heterogeneity with SGLT2i and GLP1-RA for glycaemic, cardiovascular, and renal outcomes in people with type 2 diabetes. This lack of evidence likely reflects the methodological limitations of the current evidence base. Robust future studies to fill the research gaps identified in this review are required for precision medicine in type 2 diabetes to translate to clinical care.

## Data availability

Template data collection forms and the data extracted from included studies are available upon request. All studies identified by our search protocol are detailed in Supplementary Tables 1–8.

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

## Acknowledgements

The ADA/EASD Precision Diabetes Medicine Initiative, within which this work was conducted, has received the following support: The Covidence license was funded by Lund University (Sweden) for which technical support was provided by Maria Björklund and Krister Aronsson (Faculty of Medicine Library, Lund University, Sweden). Administrative support was provided by Lund University (Malmö, Sweden), University of Chicago (IL, USA), and the American Diabetes Association (Washington D.C., USA). The Novo Nordisk Foundation (Hellerup, Denmark) provided grant support for in-person writing group meetings (PI: L Phillipson, University of Chicago, IL, USA). This study was supported by the National Institute for Health and Care Research Exeter Biomedical Research Centre. The views expressed are those of the author(s) and not necessarily those of the NIHR or the Department of Health and Social Care. K.G.Y. and J.M.D. are supported by Research England's Expanding Excellence in England (E3) fund. A.R.K. is supported by the National Center for Advancing Translational Sciences, National Institutes of Health, through Grant KL2TR002490. The content is solely the responsibility of the authors and does not necessarily represent the official views of the NIH. S.R. is funded by a US Department of Veterans Affairs Award IK2-CX001907, and a Webb-Waring Biomedical Research Award from the Boettcher Foundation. J.M.D. is funded by the EFSD Rising Star Fellowship Programme, the UK Medical Research Council (MR/N00633X/1), and a BHF-Turing Cardiovascular Data Science Award (SP/19/6/34809). M.S. is funded by the National Institutes of Health, K01HL157658. The funders had no role in the study design, data extraction or interpretation, writing of the article, or the decision to submit for publication.

## Author contributions

K.G.Y., E.H.M., R.J.M., A.R.K., S.J.P., S.R., M.A.S., D.K.T., A.P.M., A.Y.D., A.G.J., E.R.P., and J.M.D. designed the study. K.G.Y., E.H.M., R.J.M., A.R.K., S.J.P., S.R., M.A.S., A.P.M., A.Y.D., A.G.J., E.R.P., and J.M.D. implemented the systematic review and contributed to full-text data extraction. K.G.Y., E.H.M., R.J.M., A.R.K., S.J.P., S.R., M.A.S., A.Y.D., A.G.J., E.R.P., and J.M.D. synthesized the data and drafted the article. All authors critically revised the article and approved the final article. E.R.P. and J.M.D. jointly supervised this work. E.R.P. and J.M.D. attest that all listed authors meet authorship criteria, and that no others meeting the criteria have been omitted. E.R.P. and J.M.D. were responsible for the decision to submit for publication.

## Competing interests

The authors declare the following competing interests: A.P.M. declares previous research funding from Eli Lilly and Company, Pfizer, and AstraZeneca. A.G.J. has received research funding from the Novo Nordisk foundation. E.R.P. has received honoraria for speaking from Lilly, Novo Nordisk, and Illumina. All other authors declare no competing interest.

## Additional information

[1]Exeter Centre of Excellence in Diabetes (EXCEED), University of Exeter Medical School, RILD Building, Royal Devon & Exeter Hospital, Exeter, UK. [2]Division of Population Health & Genomics, School of Medicine, University of Dundee, Dundee, UK. [3]Department of Nutrition, University of North Carolina at Chapel Hill, Chapel Hill, NC, USA. [4]Department of Medicine, Johns Hopkins University School of Medicine, Baltimore, MD, USA. [5]Section of Academic Primary Care, US Department of Veterans Affairs Eastern Colorado Health Care System, Aurora, CO, USA. [6]Department of Biomedical Informatics, School of Medicine, University of Colorado, Aurora, USA. [7]Department of Nutrition, Harvard T.H. Chan School of Public Health, Boston, MA, USA. [8]Department of Medicine, Brigham and Women's Hospital, Harvard Medical School, Boston, MA, USA. [201]These authors contributed equally: Katherine G. Young, Eram Haider McInnes, Robert J. Massey. *A list of authors and their affiliations appears at the end of the paper. ✉email: e.z.pearson@dundee.ac.uk; j.dennis@exeter.ac.uk

## ADA/EASD PDMI

Deirdre K. Tobias[7,9], Jordi Merino[10,11,12], Abrar Ahmad[13], Catherine Aiken[14,15], Jamie L. Benham[16,198], Dhanasekaran Bodhini[17,198], Amy L. Clark[18], Kevin Colclough[19], Rosa Corcoy[20,21,22], Sara J. Cromer[11,23,24], Daisy Duan[25], Jamie L. Felton[26,27,28], Ellen C. Francis[29], Pieter Gillard[30], Véronique Gingras[31,32], Romy Gaillard[33], Eram Haider[34], Alice Hughes[19], Jennifer M. Ikle[35,36], Laura M. Jacobsen[37], Anna R. Kahkoska[3], Jarno L. T. Kettunen[38,39,40], Raymond J. Kreienkamp[11,12,23,41], Lee-Ling Lim[42,43,44], Jonna M. E. Männistö[45,46], Robert Massey[34], Niamh-Maire Mclennan[47], Rachel G. Miller[48], Mario Luca Morieri[49,50], Jasper Most[51], Rochelle N. Naylor[52], Bige Ozkan[53,54], Kashyap Amratlal Patel[19], Scott J. Pilla[55,56], Katsiaryna Prystupa[57,58], Sridaran Raghaven[59,60], Mary R. Rooney[53,61], Martin Schön[57,58,62], Zhila Semnani-Azad[7], Magdalena Sevilla-Gonzalez[23,24,63], Pernille Svalastoga[64,65], Wubet Worku Takele[66], Claudia Ha-ting Tam[44,67,68], Anne Cathrine B. Thuesen[10], Mustafa Tosur[69,70,71], Amelia S. Wallace[53,61], Caroline C. Wang[61], Jessie J. Wong[72], Jennifer M. Yamamoto[73], Katherine Young[19], Chloé Amouyal[74,75], Mette K. Andersen[10], Maxine P. Bonham[76], Mingling Chen[77], Feifei Cheng[78], Tinashe Chikowore[24,79,80,81], Sian C. Chivers[82], Christoffer Clemmensen[10], Dana Dabelea[83], Adem Y. Dawed[34], Aaron J. Deutsch[12,23,24], Laura T. Dickens[84], Linda A. DiMeglio[26,27,28,85], Monika Dudenhöffer-Pfeifer[13], Carmella Evans-Molina[26,27,28,86], María Mercè Fernández-Balsells[87,88], Hugo Fitipaldi[13], Stephanie L. Fitzpatrick[89], Stephen E. Gitelman[90], Mark O. Goodarzi[91,92], Jessica A. Grieger[93,94], Marta Guasch-Ferré[7,95], Nahal Habibi[93,94], Torben Hansen[10], Chuiguo Huang[44,67], Arianna Harris-Kawano[26,27,28], Heba M. Ismail[26,27,28], Benjamin Hoag[96,97], Randi K. Johnson[98,99], Angus G. Jones[19,100], Robert W. Koivula[101], Aaron Leong[11,24,102], Gloria K. W. Leung[76], Ingrid M. Libman[103], Kai Liu[93], S. Alice Long[104], William L. Lowe Jr.[105], Robert W. Morton[106,107,108],

Ayesha A. Motala[109], Suna Onengut-Gumuscu[110], James S. Pankow[111], Maleesa Pathirana[93,94], Sofia Pazmino[112], Dianna Perez[26,27,28], John R. Petrie[113], Camille E. Powe[11,23,24,114], Alejandra Quinteros[93], Rashmi Jain[115,116], Debashree Ray[61,117], Mathias Ried-Larsen[118,119], Zeb Saeed[120], Vanessa Santhakumar[9], Sarah Kanbour[55,121], Sudipa Sarkar[55], Gabriela S. F. Monaco[26,27,28], Denise M. Scholtens[122], Elizabeth Selvin[53,61], Wayne Huey-Herng Sheu[123,124,125], Cate Speake[126], Maggie A. Stanislawski[98], Nele Steenackers[112], Andrea K. Steck[127], Norbert Stefan[58,128,129], Julie Støy[130], Rachael Taylor[131], Sok Cin Tye[132,133], Gebresilasea Gendisha Ukke[66], Marzhan Urazbayeva[70,134], Bart Van der Schueren[112,135], Camille Vatier[136,137], John M. Wentworth[138,139,140], Wesley Hannah[141,142], Sara L. White[82,143], Gechang Yu[44,67], Yingchai Zhang[44,67], Shao J. Zhou[94,144], Jacques Beltrand[145,146], Michel Polak[145,146], Ingvild Aukrust[64,147], Elisa de Franco[19], Sarah E. Flanagan[19], Kristin A. Maloney[148], Andrew McGovern[19], Janne Molnes[64,147], Mariam Nakabuye[10], Pål Rasmus Njølstad[64,65], Hugo Pomares-Millan[13,149], Michele Provenzano[150], Cécile Saint-Martin[151], Cuilin Zhang[152,153], Yeyi Zhu[154,155], Sungyoung Auh[156], Russell de Souza[107,157], Andrea J. Fawcett[158,159], Chandra Gruber[160], Eskedar Getie Mekonnen[161,162], Emily Mixter[163], Diana Sherifali[107,164], Robert H. Eckel[165], John J. Nolan[166,167], Louis H. Philipson[163], Rebecca J. Brown[156], Liana K. Billings[168,169], Kristen Boyle[83], Tina Costacou[48], John M. Dennis[19], Jose C. Florez[11,12,23,24], Anna L. Gloyn[35,36,170], Maria F. Gomez[13,171], Peter A. Gottlieb[127], Siri Atma W. Greeley[172], Kurt Griffin[116,173], Andrew T. Hattersley[19,100], Irl B. Hirsch[174], Marie-France Hivert[11,175,176], Korey K. Hood[72], Jami L. Josefson[158], Soo Heon Kwak[177], Lori M. Laffel[178], Siew S. Lim[66], Ruth J. F. Loos[10,179], Ronald C. W. Ma[44,67,68], Chantal Mathieu[30], Nestoras Mathioudakis[55], James B. Meigs[24,102,180], Shivani Misra[181,182], Viswanathan Mohan[183], Rinki Murphy[184,185,186], Richard Oram[19,100], Katharine R. Owen[101,187], Susan E. Ozanne[188], Ewan R. Pearson[34], Wei Perng[83], Toni I. Pollin[148,189], Rodica Pop-Busui[190], Richard E. Pratley[191], Leanne M. Redman[192], Maria J. Redondo[69,70], Rebecca M. Reynolds[47], Robert K. Semple[47,193], Jennifer L. Sherr[194], Emily K. Sims[26,27,28], Arianne Sweeting[195,196], Tiinamaija Tuomi[38,139,40], Miriam S. Udler[11,12,23,24], Kimberly K. Vesco[197], Tina Vilsbøll[198,199], Robert Wagner[57,58,200], Stephen S. Rich[110] & Paul W. Franks[7,13,101,108]

[9]Division of Preventative Medicine, Department of Medicine, Brigham and Women's Hospital and Harvard Medical School, Boston, MA, USA. [10]Novo Nordisk Foundation Center for Basic Metabolic Research, Faculty of Health and Medical Sciences, University of Copenhagen, Copenhagen, Denmark. [11]Diabetes Unit, Endocrine Division, Massachusetts General Hospital, Boston, MA, USA. [12]Center for Genomic Medicine, Massachusetts General Hospital, Boston, MA, USA. [13]Department of Clinical Sciences, Lund University Diabetes Centre, Lund University Malmö, Sweden. [14]Department of Obstetrics and Gynaecology, the Rosie Hospital, Cambridge, UK. [15]NIHR Cambridge Biomedical Research Centre, University of Cambridge, Cambridge, UK. [16]Departments of Medicine and Community Health Sciences, Cumming School of Medicine, University of Calgary, Calgary, AB, Canada. [17]Department of Molecular Genetics, Madras Diabetes Research Foundation, Chennai, India. [18]Division of Pediatric Endocrinology, Department of Pediatrics, Saint Louis University School of Medicine, SSM Health Cardinal Glennon Children's Hospital, St. Louis, MO, USA. [19]Department of Clinical and Biomedical Sciences, University of Exeter Medical School, Exeter, UK. [20]CIBER-BBN, ISCIII, Madrid, Spain. [21]Institut d'Investigació Biomèdica Sant Pau (IIB SANT PAU), Barcelona, Spain. [22]Departament de Medicina, Universitat Autònoma de Barcelona, Bellaterra, Spain. [23]Programs in Metabolism and Medical & Population Genetics, Broad Institute, Cambridge, MA, USA. [24]Department of Medicine, Harvard Medical School, Boston, MA, USA. [25]Division of Endocrinology, Diabetes and Metabolism, Johns Hopkins University School of Medicine, Baltimore, MD, USA. [26]Department of Pediatrics, Indiana University School of Medicine, Indianapolis, IN, USA. [27]Herman B Wells Center for Pediatric Research, Indiana University School of Medicine, Indianapolis, IN, USA. [28]Center for Diabetes and Metabolic Diseases, Indiana University School of Medicine, Indianapolis, IN, USA. [29]Department of Biostatistics and Epidemiology, Rutgers School of Public Health, Piscataway, NJ, USA. [30]University Hospital Leuven, Leuven, Belgium. [31]Department of Nutrition, Université de Montréal, Montreal, Quebec, Canada. [32]Research Center, Sainte-Justine University Hospital Center, Montreal, Quebec, Canada. [33]Department of Pediatrics, Erasmus Medical Center, Rotterdam, The Netherlands. [34]Division of Population Health & Genomics, School of Medicine, University of Dundee, Dundee, UK. [35]Department of Pediatrics, Stanford School of Medicine, Stanford University, Stanford, CA, USA. [36]Stanford Diabetes Research Center, Stanford School of Medicine, Stanford University, Stanford, CA, USA. [37]University of Florida, Gainesville, FL, USA. [38]Helsinki University Hospital, Abdominal Centre/Endocrinology, Helsinki, Finland. [39]Folkhalsan Research Center, Helsinki, Finland. [40]Institute for Molecular Medicine Finland FIMM, University of Helsinki, Helsinki, Finland. [41]Department of Pediatrics, Division of Endocrinology, Boston Children's Hospital, Boston, MA, USA. [42]Department of Medicine, Faculty of Medicine, University of Malaya, Kuala Lumpur, Malaysia. [43]Asia Diabetes Foundation, Hong Kong SAR, China. [44]Department of Medicine & Therapeutics, Chinese University of Hong Kong, Hong Kong SAR, China. [45]Departments of Pediatrics and Clinical Genetics, Kuopio University Hospital, Kuopio, Finland. [46]Department of Medicine, University of Eastern Finland, Kuopio, Finland. [47]Centre for Cardiovascular Science, Queen's Medical Research Institute, University of Edinburgh, Edinburgh, UK. [48]Department of Epidemiology, University of Pittsburgh, Pittsburgh, PA, USA. [49]Metabolic Disease Unit, University Hospital of Padova, Padova, Italy. [50]Department of Medicine, University of Padova, Padova, Italy. [51]Department of Orthopedics, Zuyderland Medical Center, Sittard-Geleen, The Netherlands. [52]Departments of Pediatrics and Medicine, University of Chicago, Chicago, IL, USA. [53]Welch Center for Prevention, Epidemiology, and Clinical Research, Johns Hopkins Bloomberg School of Public

Health, Baltimore, MD, USA. [54]Ciccarone Center for the Prevention of Cardiovascular Disease, Johns Hopkins School of Medicine, Baltimore, MD, USA. [55]Department of Medicine, Johns Hopkins University, Baltimore, MD, USA. [56]Department of Health Policy and Management, Johns Hopkins University Bloomberg School of Public Health, Baltimore, MD, USA. [57]Institute for Clinical Diabetology, German Diabetes Center, Leibniz Center for Diabetes Research at Heinrich Heine University Düsseldorf, Düsseldorf, Germany. [58]German Center for Diabetes Research (DZD), Neuherberg, Germany. [59]Section of Academic Primary Care, US Department of Veterans Affairs Eastern Colorado Health Care System, Aurora, CO, USA. [60]Department of Medicine, University of Colorado School of Medicine, Aurora, CO, USA. [61]Department of Epidemiology, Johns Hopkins Bloomberg School of Public Health, Baltimore, MD, USA. [62]Institute of Experimental Endocrinology, Biomedical Research Center, Slovak Academy of Sciences, Bratislava, Slovakia. [63]Clinical and Translational Epidemiology Unit, Massachusetts General Hospital, Boston, MA, USA. [64]Mohn Center for Diabetes Precision Medicine, Department of Clinical Science, University of Bergen, Bergen, Norway. [65]Children and Youth Clinic, Haukeland University Hospital, Bergen, Norway. [66]Eastern Health Clinical School, Monash University, Melbourne, VIC, Australia. [67]Laboratory for Molecular Epidemiology in Diabetes, Li Ka Shing Institute of Health Sciences, The Chinese University of Hong Kong, Hong Kong, China. [68]Hong Kong Institute of Diabetes and Obesity, The Chinese University of Hong Kong, Hong Kong, China. [69]Department of Pediatrics, Baylor College of Medicine, Houston, TX, USA. [70]Division of Pediatric Diabetes and Endocrinology, Texas Children's Hospital, Houston, TX, USA. [71]Children's Nutrition Research Center, USDA/ARS, Houston, TX, USA. [72]Stanford University School of Medicine, Stanford, CA, USA. [73]Internal Medicine, University of Manitoba, Winnipeg, MB, Canada. [74]Department of Diabetology, APHP, Paris, France. [75]Sorbonne Université, INSERM, NutriOmic team, Paris, France. [76]Department of Nutrition, Dietetics and Food, Monash University, Melbourne, VIC, Australia. [77]Monash Centre for Health Research and Implementation, Monash University, Clayton, VIC, Australia. [78]Health Management Center, The Second Affiliated Hospital of Chongqing Medical University, Chongqing Medical University, Chongqing, China. [79]MRC/Wits Developmental Pathways for Health Research Unit, Department of Paediatrics, Faculty of Health Sciences, University of the Witwatersrand, Johannesburg, South Africa. [80]Channing Division of Network Medicine, Brigham and Women's Hospital, Boston, MA, USA. [81]Sydney Brenner Institute for Molecular Bioscience, Faculty of Health Sciences, University of the Witwatersrand, Johannesburg, South Africa. [82]Department of Women and Children's health, King's College London, London, UK. [83]Lifecourse Epidemiology of Adiposity and Diabetes (LEAD) Center, University of Colorado Anschutz Medical Campus, Aurora, CO, USA. [84]Section of Adult and Pediatric Endocrinology, Diabetes and Metabolism, Kovler Diabetes Center, University of Chicago, Chicago, USA. [85]Department of Pediatrics, Riley Hospital for Children, Indiana University School of Medicine, Indianapolis, IN, USA. [86]Richard L. Roudebush VAMC, Indianapolis, IN, USA. [87]Biomedical Research Institute Girona, IdIBGi, Girona, Spain. [88]Diabetes, Endocrinology and Nutrition Unit Girona, University Hospital Dr Josep Trueta, Girona, Spain. [89]Institute of Health System Science, Feinstein Institutes for Medical Research, Northwell Health, Manhasset, NY, USA. [90]University of California at San Francisco, Department of Pediatrics, Diabetes Center, San Francisco, CA, USA. [91]Division of Endocrinology, Diabetes and Metabolism, Cedars-Sinai Medical Center, Los Angeles, CA, USA. [92]Department of Medicine, Cedars-Sinai Medical Center, Los Angeles, CA, USA. [93]Adelaide Medical School, Faculty of Health and Medical Sciences, The University of Adelaide, Adelaide, Australia. [94]Robinson Research Institute, The University of Adelaide, Adelaide, Australia. [95]Department of Public Health and Novo Nordisk Foundation Center for Basic Metabolic Research, Faculty of Health and Medical Sciences, University of Copenhagen, 1014 Copenhagen, Denmark. [96]Division of Endocrinology and Diabetes, Department of Pediatrics, Sanford Children's Hospital, Sioux Falls, SD, USA. [97]University of South Dakota School of Medicine, Vermillion, SD, USA. [98]Department of Biomedical Informatics, University of Colorado Anschutz Medical Campus, Aurora, CO, USA. [99]Department of Epidemiology, Colorado School of Public Health, Aurora, CO, USA. [100]Royal Devon University Healthcare NHS Foundation Trust, Exeter, UK. [101]Oxford Centre for Diabetes, Endocrinology and Metabolism, University of Oxford, Oxford, UK. [102]Division of General Internal Medicine, Massachusetts General Hospital, Boston, MA, USA. [103]UPMC Children's Hospital of Pittsburgh, Pittsburgh, PA, USA. [104]Center for Translational Immunology, Benaroya Research Institute, Seattle, WA, USA. [105]Department of Medicine, Northwestern University Feinberg School of Medicine, Chicago, IL, USA. [106]Department of Pathology & Molecular Medicine, McMaster University, Hamilton, Canada. [107]Population Health Research Institute, Hamilton, Canada. [108]Department of Translational Medicine, Medical Science, Novo Nordisk Foundation, Hellerup, Denmark. [109]Department of Diabetes and Endocrinology, Nelson R Mandela School of Medicine, University of KwaZulu-Natal, Durban, South Africa. [110]Center for Public Health Genomics, Department of Public Health Sciences, University of Virginia, Charlottesville, VA, USA. [111]Division of Epidemiology and Community Health, School of Public Health, University of Minnesota, Minneapolis, MN, USA. [112]Department of Chronic Diseases and Metabolism, Clinical and Experimental Endocrinology, KU Leuven, Leuven, Belgium. [113]School of Health and Wellbeing, College of Medical, Veterinary and Life Sciences, University of Glasgow, Glasgow, UK. [114]Department of Obstetrics, Gynecology, and Reproductive Biology, Massachusetts General Hospital and Harvard Medical School, Boston, MA, USA. [115]Sanford Children's Specialty Clinic, Sioux Falls, SD, USA. [116]Department of Pediatrics, Sanford School of Medicine, University of South Dakota, Sioux Falls, SD, USA. [117]Department of Biostatistics, Johns Hopkins Bloomberg School of Public Health, Baltimore, MD, USA. [118]Centre for Physical Activity Research, Rigshospitalet, Copenhagen, Denmark. [119]Institute for Sports and Clinical Biomechanics, University of Southern Denmark, Odense, Denmark. [120]Department of Medicine, Division of Endocrinology, Diabetes and Metabolism, Indiana University School of Medicine, Indianapolis, IN, USA. [121]AMAN Hospital, Doha, Qatar. [122]Department of Preventive Medicine, Division of Biostatistics, Northwestern University Feinberg School of Medicine, Chicago, IL, USA. [123]Institute of Molecular and Genomic Medicine, National Health Research Institutes, Taipei City, Taiwan. [124]Divsion of Endocrinology and Metabolism, Taichung Veterans General Hospital, Taichung, Taiwan. [125]Division of Endocrinology and Metabolism, Taipei Veterans General Hospital, Taipei, Taiwan. [126]Center for Interventional Immunology, Benaroya Research Institute, Seattle, WA, USA. [127]Barbara Davis Center for Diabetes, University of Colorado Anschutz Medical Campus, Aurora, CO, USA. [128]University Hospital of Tübingen, Tübingen, Germany. [129]Institute of Diabetes Research and Metabolic Diseases (IDM), Helmholtz Center Munich, Neuherberg, Germany. [130]Steno Diabetes Center Aarhus, Aarhus University Hospital, Aarhus, Denmark. [131]University of Newcastle, Newcastle upon Tyne, UK. [132]Sections on Genetics and Epidemiology, Joslin Diabetes Center, Harvard Medical School, Boston, MA, USA. [133]Department of Clinical Pharmacy and Pharmacology, University Medical Center Groningen, Groningen, The Netherlands. [134]Gastroenterology, Baylor College of Medicine, Houston, TX, USA. [135]Department of Endocrinology, University Hospitals Leuven, Leuven, Belgium. [136]Sorbonne University, Inserm U938, Saint-Antoine Research Centre, Institute of Cardiometabolism and Nutrition, Paris, France. [137]Department of Endocrinology, Diabetology and Reproductive Endocrinology, Assistance Publique-Hôpitaux de Paris, Saint-Antoine University Hospital, National Reference Center for Rare Diseases of Insulin Secretion and Insulin Sensitivity (PRISIS), Paris, France. [138]Royal Melbourne Hospital Department of Diabetes and Endocrinology, Parkville, VIC, Australia. [139]Walter and Eliza Hall Institute, Parkville, VIC, Australia. [140]University of Melbourne Department of Medicine, Parkville, VIC, Australia. [141]Deakin University, Melbourne, Australia. [142]Department of Epidemiology, Madras Diabetes Research Foundation, Chennai, India. [143]Department of Diabetes and Endocrinology, Guy's and St Thomas' Hospitals NHS Foundation Trust, London, UK. [144]School of Agriculture, Food and Wine, University of Adelaide, Adelaide, Australia. [145]Institut Cochin, Paris, France. [146]Pediatric endocrinology and diabetes, Hopital Necker Enfants Malades, APHP Centre, université de Paris, Paris, France. [147]Department of Medical Genetics, Haukeland University Hospital, Bergen, Norway. [148]Department of Medicine, University of Maryland School of Medicine, Baltimore, MD, USA. [149]Department of Epidemiology, Geisel School of Medicine at Dartmouth, Hanover, NH, USA. [150]Nephrology, Dialysis and Renal Transplant Unit, IRCCS—Azienda Ospedaliero-Universitaria di

Bologna, Alma Mater Studiorum University of Bologna, Bologna, Italy. [151]Department of Medical Genetics, AP-HP Pitié-Salpêtrière Hospital, Sorbonne University, Paris, France. [152]Global Center for Asian Women's Health, Yong Loo Lin School of Medicine, National University of Singapore, Singapore, Singapore. [153]Department of Obstetrics and Gynecology, Yong Loo Lin School of Medicine, National University of Singapore, Singapore, Singapore. [154]Kaiser Permanente Northern California Division of Research, Oakland, CA, USA. [155]Department of Epidemiology and Biostatistics, University of California San Francisco, California, USA. [156]National Institute of Diabetes and Digestive and Kidney Diseases, National Institutes of Health, Bethesda, MD, USA. [157]Department of Health Research Methods, Evidence, and Impact, Faculty of Health Sciences, McMaster University, Hamilton, ON, Canada. [158]Ann & Robert H. Lurie Children's Hospital of Chicago, Department of Pediatrics, Northwestern University Feinberg School of Medicine, Chicago, IL, USA. [159]Department of Clinical and Organizational Development, Chicago, IL, USA. [160]American Diabetes Association, Arlington, Virginia, USA. [161]College of Medicine and Health Sciences, University of Gondar, Gondar, Ethiopia. [162]Global Health Institute, Faculty of Medicine and Health Sciences, University of Antwerp, Antwerp, Belgium. [163]Department of Medicine and Kovler Diabetes Center, University of Chicago, Chicago, IL, USA. [164]School of Nursing, Faculty of Health Sciences, McMaster University, Hamilton, Canada. [165]Division of Endocrinology, Metabolism, Diabetes, University of Colorado, Boulder, CO, USA. [166]Department of Clinical Medicine, School of Medicine, Trinity College Dublin, Dublin, Ireland. [167]Department of Endocrinology, Wexford General Hospital, Wexford, Ireland. [168]Division of Endocrinology, NorthShore University HealthSystem, Skokie, IL, USA. [169]Department of Medicine, Prtizker School of Medicine, University of Chicago, Chicago, IL, USA. [170]Department of Genetics, Stanford School of Medicine, Stanford University, Stanford, CA, USA. [171]Faculty of Health, Aarhus University, Aarhus, Denmark. [172]Departments of Pediatrics and Medicine and Kovler Diabetes Center, University of Chicago, Chicago, USA. [173]Sanford Research, Sioux Falls, SD, USA. [174]University of Washington School of Medicine, Seattle, WA, USA. [175]Department of Population Medicine, Harvard Medical School, Harvard Pilgrim Health Care Institute, Boston, MA, USA. [176]Department of Medicine, Universite de Sherbrooke, Sherbrooke, QC, Canada. [177]Department of Internal Medicine, Seoul National University College of Medicine, Seoul National University Hospital, Seoul, Republic of Korea. [178]Joslin Diabetes Center, Harvard Medical School, Boston, MA, USA. [179]Charles Bronfman Institute for Personalized Medicine, Icahn School of Medicine at Mount Sinai, New York, NY, USA. [180]Broad Institute, Cambridge, MA, USA. [181]Division of Metabolism, Digestion and Reproduction, Imperial College London, London, UK. [182]Department of Diabetes & Endocrinology, Imperial College Healthcare NHS Trust, London, UK. [183]Department of Diabetology, Madras Diabetes Research Foundation & Dr. Mohan's Diabetes Specialities Centre, Chennai, India. [184]Department of Medicine, Faculty of Medicine and Health Sciences, University of Auckland, Auckland, New Zealand. [185]Auckland Diabetes Centre, Te Whatu Ora Health New Zealand, Auckland, New Zealand. [186]Medical Bariatric Service, Te Whatu Ora Counties, Health New Zealand, Auckland, New Zealand. [187]Oxford NIHR Biomedical Research Centre, University of Oxford, Oxford, UK. [188]University of Cambridge, Metabolic Research Laboratories and MRC Metabolic Diseases Unit, Wellcome-MRC Institute of Metabolic Science, Cambridge, UK. [189]Department of Epidemiology & Public Health, University of Maryland School of Medicine, Baltimore, MD, USA. [190]Department of Internal Medicine, Division of Metabolism, Endocrinology and Diabetes, University of Michigan, Ann Arbor, MI, USA. [191]AdventHealth Translational Research Institute, Orlando, FL, USA. [192]Pennington Biomedical Research Center, Baton Rouge, LA, USA. [193]MRC Human Genetics Unit, Institute of Genetics and Cancer, University of Edinburgh, Edinburgh, UK. [194]Yale School of Medicine, New Haven, CT, USA. [195]Faculty of Medicine and Health, University of Sydney, Sydney, NSW, Australia. [196]Department of Endocrinology, Royal Prince Alfred Hospital, Sydney, NSW, Australia. [197]Kaiser Permanente Northwest, Kaiser Permanente Center for Health Research, Portland, OR, USA. [198]Clinial Research, Steno Diabetes Center Copenhagen, Herlev, Denmark. [199]Department of Clinical Medicine, Faculty of Health and Medical Sciences, University of Copenhagen, Copenhagen, Denmark. [200]Department of Endocrinology and Diabetology, University Hospital Düsseldorf, Heinrich Heine University Düsseldorf, Düsseldorf, Germany.

