## [Peer Review File · Communications Medicine]

Reviewers' comments:

Reviewer #1 (Remarks to the Author):

Precision medicine facilitates individualised drug choice for people living with type 2 diabetes. A number of research groups aim at it through different strategies. This systematic review comprehensively summarised the literature of the field qualitatively. It is undoubtedly important for its topic. This study also summarised a number of research gaps highlighting what needs mostly at the current stage.

Generally, an umbrella review may better fit the study propose. [ED: We appreciate that you cannot change approach; perhaps this can be suggested for future studies?] The quantitative comparison of findings between randomised trials and observational studies are quite interests.

Technical comments

Line 140, why 4 months for eligibility?

Line 174-175, the study quoted only the JBI tool for cohorts. What about the randomised trials?

Line 178-179, great to find the adoption of GRADE. But GRADE assesses the credibility of the evidence as high, moderate, low and very low. It is better to show the details transparently.

Line 182, the outcomes of interest could move to the eligibility criteria.

Line 183-184, what does 'changes in HbA1c' mean? Does it mean 'HbA1c change from baseline', 'mean difference between groups', or 'the last follow-up HbA1c'?

Line 187, why not ESKD or creatinine doubling?

Table 1, good to know the effect sizes if available.

Reviewer #2 (Remarks to the Author):

This is one of the most comprehensive state-of-art reviews for the treatment effect heterogeneity for GLP-1RA and SGLT2i. The authors performed a thorough literature digging and comparative study in terms of a wide range of demographic, clinical and biological features, including pharmacogenetic markers, to assess evidence for treatment effect modification for these two drug classes. They found that lower renal function as a predictor of lesser glycaemic response with SGLT2i, while markers of reduced insulin secretion were identified as predictors of lesser glycaemic response with GLP1-receptor agonists. However, they did not identify clear factors that alter heart and renal disease outcomes for either treatment. So far, the evidence of treatment effect heterogeneity with SGLT2i and GLP1-RA is limited, likely reflecting the methodological limitations of published studies. The manuscript highlighted the great gap for practicing precision medicine of SGLT2i and GLP-1RA with current evidence. Although it did not provide so much ensured insights but sometimes raising the questions might be more important than answering the questions. Generally speaking, the manuscript is within the scope of the journal. The standard and rigorous systematic review protocol and evidence quality assessment tools have been adopted. The summary of the evidence is clear and reader-friendly. It is a well-written manuscript but I still have several suggestions for the further improvement.

1. According to the supplementary table 1 and table 2, I think there are some important CVOTs of SGLT2i and GLP-1RA not included in the evidence integration. For SGLT2i, DELIVER study (for HFpEF) and EMPA-Kidney should be included. For GLP-1RA, AMPLITUDE-O study should be included. Please update the analysis with a more comprehensive study inclusion.

2. Although the authors summarized that lower renal function as a predictor of lesser glycaemic response with SGLT2i, while markers of reduced insulin secretion were identified as predictors of lesser glycaemic response with GLP1-receptor agonists, these findings were only applied within the same drug classes. Without direct or indirect comparisons, no clues of treatment effect heterogeneity between SGLT2i versus GLP-1RA were revealed by the systematic review. It is really a big gap for individualized medicine.

3. As for the cardiovascular and renal outcomes of SGLT2i and GLP-1RA, previous researches have indicated the benefits heterogeneity between SGLT2i and GLP-1RA. The anti-atherosclerotic benefits might be predominant in GLP-1RA while the “pump” benefits for heart failure and renal failure is more prominent in SGLT2i. Unlike SGLT2i, the GLP-1RA might be effective in improving the albuminuria but the solid evidence for reducing the renal events is still lacking. This therapeutic heterogeneity is also helpful for precision medicine decision making, which was not adequately discussed in the manuscript.

4. There are some publications that might further add to the evidence summary. I would like to provide the relevant information here for the authors’ reference.

(1) Ma Y, Lin C, Cai X, Hu S, Zhu X, Lv F, Yang W, Ji L. Baseline eGFR, albuminuria and renal outcomes in patients with SGLT2 inhibitor treatment: an updated meta-analysis. *Acta Diabetol.* 2023 Mar;60(3):435-445.

(2) Hu S, Lin C, Cai X, Zhu X, Lv F, Yang W, Ji L. Disparities in efficacy and safety of sodium-glucose cotransporter 2 inhibitor among patients with different extents of renal dysfunction: A systematic review and meta-analysis of randomized controlled trials. *Front Pharmacol.* 2022 Nov 22;13:1018720.

(3) Lin C, Cai X, Ji L. Cardiovascular benefits beyond urinary glucose excretion: A hypothesis generated from two meta-analyses. *Diabetes Obes Metab.* 2022 Mar;24(3):550-554.

(4) Hu M, Cai X, Yang W, Zhang S, Nie L, Ji L. Effect of Hemoglobin A1c Reduction or Weight Reduction on Blood Pressure in Glucagon-Like Peptide-1 Receptor Agonist and Sodium-Glucose Cotransporter-2 Inhibitor Treatment in Type 2 Diabetes Mellitus: A Meta-Analysis. *J Am Heart Assoc.* 2020 Apr 7;9(7):e015323.

Reviewer #3 (Remarks to the Author):

Young and colleagues performed a systematic review of 101 SGLT2 inhibitor studies and 75 GLP1 receptor agonist studies to evaluate variability in treatment response to these relatively new therapies for the treatment of diabetes. They determined response variability in terms of glycemic responses and clinical cardiovascular and kidney outcomes. Their main conclusions are that kidney function determined the heterogeneity in glycemic response to SGLT2 inhibitors and markers of insulin resistance determined the heterogeneity in response to GLP-1RA. No other clinical characteristics or biomarkers were reliably predicting the response to these agents.

1) Overall the literature search and the data extraction is well performed. In addition the topic is timely and relevant. However, the novelty is extremely limited. It is already known for nearly a decade that a lower kidney function (Glomerular Filtration Rate) is associated with a poorer glycemic response to SGLT2 inhibitors just like insulin resistance being a determinant of response to GLP-1 RA. In addition, many studies from the outcome trials have shown that the responses in terms of clinical outcomes are consistent across a range of patient characteristics. This study confirms these findings but does not add anything new.

2) I believe that the observational studies should be removed from this report. Variation in glycemic response in observational studies are highly likely reflecting regression to the mean (patients with higher baseline Hba1c show a larger reduction at the follow-up visit just because of regression to the mean). I would recommend that the authors focus on clinical trials.

[ED: Please do not remove these studies, you could instead discuss the limitations of these studies and separate out your assessment of the different types of study?]

3) Instead of concluding that there is no response variation (that should be concluded from the findings) the authors state that we need larger studies to determine response variation. How large should these studies be if we cannot detect a signal in this large systematic review? They also state that in the future we can better predict individual treatment response, on what basis do the authors believe we can predict individual treatment response in the future? Additional explanations would help.

Response to reviewers for: Precision medicine in type 2 diabetes: A systematic review of treatment effect heterogeneity for GLP1-receptor agonists and SGLT2-inhibitors (COMMSMED-23-0310)

Reviewer 1 comments	Author response	Changes made (manuscript changes in italics; line numbers are those in the revised manuscript)
Precision medicine facilitates individualised drug choice for people living with type 2 diabetes. A number of research groups aim at it through different strategies. This systematic review comprehensively summarised the literature of the field qualitatively. It is undoubtedly important for its topic. This study also summarised a number of research gaps highlighting what needs mostly at the current stage. The quantitative comparison of findings between randomised trials and observational studies are quite interesting.	We thank the reviewer for their comments and for taking the time to review our study.	-
Generally, an umbrella review may better fit the study proposal. [ED: We appreciate that you cannot change approach; perhaps this can be suggested for future studies?]	As detailed in our pre-specified analysis protocol (PROSPERO submission CRD42022303236), we included a broad range of studies including primary studies, systematic reviews and meta-analysis. We agree that an umbrella review, which could include types of review that we did not consider such as narrative reviews, could provide further insight into the current state-of-play and research gaps in precision Type 2 diabetes treatment, and have now included this in our discussion.	Text added after line 621 in the discussion: "In addition, different methods for synthesising the current available evidence, such as conducting an umbrella review, may offer further insights into the current state-of-play of precision Type 2 diabetes treatment."
Line 140, why 4 months for eligibility?	4 months (or 16 weeks) was pre-specified pragmatically as the minimum follow-up time as this is commonly used in glycaemia trials as a suitable period over which changes in HbA1c from baseline resulting from treatment can be assessed. We have added this detail to the methods.	Line 141 now reads: "At least a 4-month potential follow up period (chosen pragmatically as a suitable time length over which changes in glycaemic could be assessed) after initiation of the drug class of interest."
Line 174-175, the study quoted only the JBI tool for cohorts. What about the randomised trials?	We apologise that this was not clear; we used the JBI Cohort Studies tool for all studies (observational and RCTs) to allow risk of bias within each drug-outcome category to be evaluated. We have amended the text on	Lines 177-178 now reads: "Risk of bias evaluations were conducted alongside the data extraction by each pair of authors, using the Joanna Briggs

	lines 174-175 to make this clear. We feel the Cohort studies tool is most appropriate for the included trial studies as we only identified and included post-hoc analysis of RCTs that examine treatment effect heterogeneity, and we did not identify any RCTs specifically designed to test treatment effect heterogeneity.	Institute (JBI) Critical Appraisal Tool for Cohort Studies¹⁰ for all included research papers
Line 178-179, great to find the adoption of GRADE. But GRADE assesses the credibility of the evidence as high, moderate, low and very low. It is better to show the details transparently.	Thank you, we have added to the 'Summary of quality assessment' paragraph (lines 495-508 in the original manuscript) to detail the credibility of evidence that each GRADE score relates to.	Line 505-506: "GRADE B: moderate certainty: the true effect is probably close to the estimated effect " Line 511: "GRADE C (low certainty: the true effect might be markedly different from the estimated effect)"
Line 182, the outcomes of interest could move to the eligibility criteria.	We think for clarity, the details of the study outcomes are best included in a separate section; however, we have added a reference to this section in the inclusion criteria to signpost readers to where they can find these details. We are happy to amend further if the editor wishes.	Line 149: "Reported differential effects of the drug class on an outcome of interest (see Outcomes section below) with respect to a biomarker."
Line 183-184, what does 'changes in HbA1c' mean? Does it mean 'HbA1c change from baseline', 'mean difference between groups', or 'the last follow-up HbA1c'?	Apologies that this was not clear, we have amended to clarify that this was change in HbA1c from baseline.	Line 186: "changes in HbA1c from baseline associated with treatment"
Line 187, why not ESKD or creatinine doubling?	Our renal outcomes did include ESKD as part of 'development of chronic kidney disease', and included changes in creatinine under 'changes in eGFR'. We have amended the text to clarify this.	Lines 191-192: "renal outcomes including development of chronic kidney disease (including end-stage renal disease, ESRD), and longitudinal changes in markers of renal function including eGFR/ creatinine and albuminuria."
Table 1, good to know the effect sizes if available.	Unfortunately as there is such heterogeneity between the studies in terms of the subgroup and outcome definitions and analysis methods, to include the effect sizes for all studies in a meaningful way we would need to include a great deal of detail from each study to provide context. This would not allow the collective evidence for each of our outcomes to be easily	Text added after line 618 in the discussion: "As our aim was to provide a comprehensive review of these treatments, we did not conduct quantitative analysis of specific biomarkers due to the range of different biomarkers, methodologies, and outcomes evaluated in the included studies. However, this review provides guidance for where

	synthesized and assessed by the reader, which is the aim of Table 1. We have included the effect sizes in the main body of the text for the studies of most interest for each outcome, including the main trials and trial meta-analyses for cardiovascular and kidney outcomes, and the larger glycaemic outcome studies. We acknowledge this is a limitation of our review, and have now highlighted this in the Discussion. In our view, more targeted systematic review / meta-analysis studies (for example focusing on effect modification associated with a specific biomarker and/or outcome) are needed in this area.	future targeted quantitative meta-analysis could be most insightful."
--	--	---

Reviewer 2 comments	Author response	Changes made (manuscript changes in italics; line numbers are those in original manuscript)
This is one of the most comprehensive state-of-art reviews for the treatment effect heterogeneity for GLP-1RA and SGLT2i. The authors performed a thorough literature digging and comparative study in terms of a wide range of demographic, clinical and biological features, including pharmacogenetic markers, to assessed evidence for treatment effect modification for these two drug classes. They found that lower renal function as a predictor of lesser glycaemic response with SGLT2i, while markers of reduced insulin secretion were identified as predictors of lesser glycaemic response with GLP1-receptor agonists. However, they did not identify clear factors that alter heart and renal disease outcomes for either treatment. So far, the evidence of treatment effect heterogeneity with SGLT2i and GLP1-RA is limited, likely reflecting the methodological limitations of published studies. The manuscript highlighted the great gap for practicing precision medicine of SGLT2i and GLP-1RA with current evidence. Although it did not provide so much ensured insights but sometimes raising the questions might be more important than answering the questions. Generally speaking, the manuscript is within the scope of the journal. The standard and rigorous systematic review protocol and evidence quality assessment tools have been adopted. The summary of the evidence is clear and reader-friendly. It is a well-written manuscript but I still have several suggestions for the further improvement.	We thank the reviewer for their comments and for taking the time to review our study and offer their suggestions for improvements.	-
1. According to the supplementary table 1 and table 2, I think there are some important CVOTs of SGLT2i and GLP-1RA not included in the evidence integration. For SGLT2i, DELIVER study (for HFpEF) and EMPA-Kidney should be included. For GLP-1RA, AMPLITUDE-O study should be included. Please update the analysis with a more comprehensive study inclusion.	Unfortunately as our literature search was conducted in February 2022, we were unable to include the DELIVER and EMPA-Kidney trials or any meta-analysis which included them, as their results weren't published until later in 2022. We agree that it is important that any new trials of SGLT2i and GLP1-RA are incorporated into future heterogeneity-focused meta-analyses and systematic reviews, and have now added a line in our discussion to this effect.	Text added lines 615-616 in the discussion: "Further SGLT2i and GLP1-RA trials are also underway and may add to the evidence presented in this review."

	Our literature search for GLP1-RA papers did not include 'efpeglenatide' (and thus did not identify the primary results paper of the AMPLITUDE-O study) as we only included licensed GLP1-RA; as far as we are aware efpeglenatide is not yet licensed. We have updated our methods to specify this. In addition, we were most interested in identifying meta-analyses rather than individual trials; as stated in our discussion (lines 554-55 of updated manuscript), "individual RCTs are by design powered only for the main effect of treatment, our primary focus when reporting were meta-analyses of post-hoc subgroup analyses of RCTs". None of the meta-analyses we identified included AMPLITUDE-O.	Test added to line 126 of the methods: "Terms for drug class (SGLT2i or GLP1-RA) and individual generic names of licensed drugs within each class (e.g. 'empagliflozin') were included."
2. Although the authors summarized that lower renal function as a predictor of lesser glycaemic response with SGLT2i, while markers of reduced insulin secretion were identified as predictors of lesser glycaemic response with GLP1-receptor agonists, these findings were only applied within the same drug classes. Without direct or indirect comparisons, no clues of treatment effect heterogeneity between SGLT2i versus GLP-1RA were revealed by the systematic review. It is really a big gap for individualized medicine.	We agree and highlight in our discussion that the most robust evidence for precision individualised treatment comes from head-to-head trials of the treatments (lines 559-565). Unfortunately, as we state in the first bullet point of our list of high-level gaps found by our review (lines 575-576), we found limited studies of this type. We believe our current discussion emphasises this important shortcoming in the current literature, and we have highlighted several studies published since our review was undertaken that do specifically compare head-to-head treatment effect heterogeneity across drug classes (notable the Trimaster RCT), although none of these studies specifically examine SGLT2i versus GLP1-RA. We are happy to expand this section of the discussion if the editor wishes.	-
3. As for the cardiovascular and renal outcomes of SGLT2i and GLP-1RA, previous researches have indicated the benefits heterogeneity between SGLT2i and GLP-1RA. The anti-atherosclerotic benefits might be predominant in GLP-1RA while the "pump" benefits for heart failure and renal failure is more prominent in SGLT2i. Unlike SGLT2i, the GLP-1RA might be	Thank you for this comment - we agree of the superiority, on average, of GLP1-RA over SGLT2i for atherosclerotic heart disease, and the superiority of SGLT2i over GLP1-RA for heart and renal failure. As noted by the reviewer, these differences reflect average treatment effects. We have tried to discuss these fully in the context of the paper - highlight these differences in	-

effective in improving the albuminuria but the solid evidence for reducing the renal events is still lacking. This therapeutic heterogeneity is also helpful for precision medicine decision making, which was not adequately discussed in the manuscript.	the Introduction (paragraph 1), including referencing relevant meta-analyses studies for each drug class (McGuire JAMA Cardiol (2021) for SGLT2i, Sattar Lancet D&E (2021) for GLP1-RA). We have not revisited this point in detail in the Discussion to ensure we maintain focus of the novelty of our study which is establishing the evidence based for treatment effect heterogeneity (i.e. moving beyond average treatment effects)	
4. There are some publications that might further add to the evidence summary. I would like to provide the relevant information here for the authors' reference. (1) Ma Y, Lin C, Cai X, Hu S, Zhu X, Lv F, Yang W, Ji L. Baseline eGFR, albuminuria and renal outcomes in patients with SGLT2 inhibitor treatment: an updated meta-analysis. Acta Diabetol. 2023 Mar;60(3):435-445. (2) Hu S, Lin C, Cai X, Zhu X, Lv F, Yang W, Ji L. Disparities in efficacy and safety of sodium-glucose cotransporter 2 inhibitor among patients with different extents of renal dysfunction: A systematic review and meta-analysis of randomized controlled trials. Front Pharmacol. 2022 Nov 22;13:1018720. (3) Lin C, Cai X, Ji L. Cardiovascular benefits beyond urinary glucose excretion: A hypothesis generated from two meta-analyses. Diabetes Obes Metab. 2022 Mar;24(3):550-554. (4) Hu M, Cai X, Yang W, Zhang S, Nie L, Ji L. Effect of Hemoglobin A1c Reduction or Weight Reduction on Blood Pressure in Glucagon-Like Peptide-1 Receptor Agonist and Sodium-Glucose Cotransporter-2 Inhibitor Treatment in Type 2 Diabetes Mellitus: A Meta-Analysis. J Am Heart Assoc. 2020 Apr 7;9(7):e015323.	We thank the reviewer for their suggestions of further literature to include. Unfortunately as our literature search was conducted in February 2022, the first 3 papers were not yet published so could not be included. Paper 4 was picked up by our literature search but excluded due to the outcome being blood pressure rather than one of the glycaemic, cardiovascular or renal outcomes in our pre-specified analysis plan.	-

Reviewer 3 comments	Author response	Changes made (manuscript changes in italics; line numbers are those in original manuscript)
Young and colleagues performed a systematic review of 101 SGLT2 inhibitor studies and 75 GLP1 receptor agonist studies to evaluate variability in treatment response to these relatively new therapies for the treatment of diabetes. They determined response variability in terms of glycemic responses and clinical cardiovascular and kidney outcomes. Their main conclusions are that kidney function determined the heterogeneity in glycemic response to SGLT2 inhibitors and markers of insulin resistance determined the heterogeneity in response to GLP-1RA. No other clinical characteristics or biomarkers were reliably predicting the response to these agents.	We would like to thank the reviewer for taking the time to consider our manuscript.	
1) Overall the literature search and the data extraction is well performed. In addition the topic is timely and relevant. However, the novelty is extremely limited. It is already known for nearly a decade that a lower kidney function (Glomerular Filtration Rate) is associated with a poorer glycemic response to SGLT2 inhibitors just like insulin resistance being a determinant of response to GLP-1 RA. In addition, many studies from the outcome trials have shown that the responses in terms of clinical outcomes are consistent across a range of patient characteristics. This study confirms these findings but does not add anything new.	We thank the reviewer for their comment and are glad your view is that the literature search and data extraction is well performed. We disagree that our review has limited novelty. The purpose and novelty of our review is to synthesise the existing evidence based on treatment effect heterogeneity for major clinical outcomes, which has not been previously done for these two major drug classes. Major novelties of our findings include: 1) we identify several biomarkers where there is robust evidence for treatment effect heterogeneity, notably (and as the reviewer highlights) for lower renal function and reduced glycaemic response to SGLT2i, and for markers of reduced insulin secretion and reduced glycaemic response to GLP1-RA. We do not feel the latter, in particular, is widely known. 2) We highlight the lack of clear evidence for treatment effect heterogeneity for other biomarkers and outcomes, and how this likely relates to the methodological robustness of existing studies. This highlights need for appropriately designed future studies of treatment effect heterogeneity,	-

	ideally directly comparing active drug classes head-to-head. As noted in the discussion, individual outcome trials are, by definition, underpowered to evaluate heterogeneity in participant subgroups, so do not in themselves provide clear evidence of an absence of treatment effect heterogeneity.	
2) I believe that the observational studies should be removed from this report. Variation in glycemic response in observational studies are highly likely reflecting regression to the mean (patients with higher baseline Hba1c show a larger reduction at the follow-up visit just because of regression to the mean). I would recommend that the authors focus on clinical trials. [ED: Please do not remove these studies, you could instead discuss the limitations of these studies and separate out your assessment of the different types of study?]	We thank the reviewer for their comment. We appreciate that glycaemic response within observational studies is less easily attributed to direct therapeutic action than in clinical trials. We have therefore added a sentence to the discussion to make this explicit. While we understand the difficulty in comparing observational and clinical trial data, we do not think separation of them in the text would be beneficial for our review of studies investigating glycaemic outcomes. This is because the findings of the observational and clinical trial studies were largely in alignment, thus discussing them together allows for easier synthesis of the results by the reader.	The following sentence has been added at line 531: “Of note, many of the included studies for HbA1c outcome were observational, meaning findings could potentially reflect biases from differential prescribing behaviour, or regression to the mean, although we did attempt to account for the latter by including adjustment for baseline HbA1c as one of our study inclusion criteria.”
3) Instead of concluding that there is no response variation (that should be concluded from the findings) the authors state that we need larger studies to determine response variation. How large should these studies be if we cannot detect a signal in this large systematic review? They also state that in the future we can better predict individual treatment response, on what basis do the authors believe we can predict individual treatment response in the future? Additional explanations would help.	We thank the reviewer for raising these important points. However, we disagree with the reviewer’s interpretation of our conclusions. Within our manuscript, we do not recommend larger studies to be performed, instead we suggest more robust studies must be performed. By this, we mean studies that are designed specifically to test for treatment heterogeneity. Indeed, we have highlighted our opinion as to the gaps in the current research with a bullet-pointed list in the discussion, and larger studies are not one of our recommendations. Regarding prediction of treatment response, we agree that it is an open question as to whether a better understanding of the covariates that influence treatment heterogeneity will help us predict patient response in the future. Hence, we do not make a conclusive statement about this in our	-

	discussion, and instead say that more research may help us in this regard in the future.	
--	--	--

Reviewers' comments:

Reviewer #1 (Remarks to the Author):

Thanks for the response from the authors in detail. They are very helpful and improve the manuscript greatly. A couple of issues remain.

1. It is apparently inappropriate to assess the RoB of RCTs using a tool for that of cohorts. My understanding from the response is that this study did not actually include any RCTs and the authors considered the post hoc analyses of RCTs (please make sure they are not predefined analyses) as cohorts. Despite not being optimal (non-randomised studies of intervention, NRSI, may be more appropriate), it is an acceptable explanation and deserves a statement in the Method section.
2. For the GRADE statements, the B and C things are not acceptable. Please refer to the standard implication for GRADE results (<https://doi.org/10.1016/j.jclinepi.2019.10.014>).

Reviewer #2 (Remarks to the Author):

Thanks for the authors' point-to-point responses to my concerns. However, not including the event-driven outcome trials such as DELIVER, EMPA-Kidney and AMPLITUDE-O is still inappropriate from my own perspective. I am aware that the literature search was conducted in February 2022 when these trials were not officially published. However, it has been more than one year since the primary literature search. The literature search should be updated at this moment. We are talking about the large-scale and well-designed CVOTs instead of certain phase 2 or phase 3 efficacy evaluation trials with limited participants. The authors now are making their efforts to produce a state-of-art summary of high-quality evidence for precision medicine in applying GLP-1RAs and SGLT2i at clinical settings. Without the latest CVOTs included, the analyses could be biased.

Reviewer #3 (Remarks to the Author):

I thank the authors for their comprehensive response to my comments. I remain of opinion that the novelty of this paper is limited and specific gaps to be addressed remain vague. However, i recognize this is subjective and respect the authors have a different opinion.

Response to reviewers for: Precision medicine in type 2 diabetes: A systematic review of treatment effect heterogeneity for GLP1-receptor agonists and SGLT2-inhibitors (COMMSMED-23-0310)

Reviewers' comments:

Reviewer #1 (Remarks to the Author):

Thanks for the response from the authors in detail. They are very helpful and improve the manuscript greatly. A couple of issues remain.

1. It is apparently inappropriate to assess the RoB of RCTs using a tool for that of cohorts. My understanding from the response is that this study did not actually include any RCTs and the authors considered the post hoc analyses of RCTs (please make sure they are not predefined analyses) as cohorts. Despite not being optimal (non-randomised studies of intervention, NRSI, may be more appropriate), it is an acceptable explanation and deserves a statement in the Method section.

>> Thank you – the reviewer is correct, and we agree that an additional statement in the Methods explaining our choice of RoB tool is warranted. We have added the following statement to the Methods:

Methods, paragraph 6 (Data Extraction and Quality Assessment): “The Cohort studies tool was applied for all studies as we did not identify any individual RCTs designed to specifically examine treatment effect heterogeneity, and all included RCT meta-analyses represent post-hoc rather than pre-specified analyses.”

2. For the GRADE statements, the B and C things are not acceptable. Please refer to the standard implication for GRADE results (<https://doi.org/10.1016/j.jclinepi.2019.10.014>).

>> Thank you for this suggestion. We have now provided our GRADE assessment as a Table in line with the suggested reference. Please see the new Table 2 in the revised manuscript. We have also updated the text reporting our GRADE assessment in the Results and Discussion (note now much more detailed reporting is provided in the accompanying table):

Summary of Quality Assessment: “Overall certainty of evidence was rated as moderate for all outcomes except glycaemia with GLP1-RA which was rated low certainty. This reflects that a larger proportion of the studies included for evaluation of GLP1-RA glycaemia outcomes were observational (24/49). By contrast, for SGLT2i glycaemia outcomes there were 18 RCT/meta-analyses and 9 observational studies. For CVD and renal outcomes, observational studies were limited and the majority of evidence came from industry-funded CVOTs (RCT designs), including post-hoc analyses of individual trials as well as meta-analyses.”

*Discussion (additions in bold): “For glycaemic response, there was **high certainty** that reduced renal function is associated with lower efficacy of SGLT2i. For GLP1-RA **there was moderate certainty that** markers of reduced insulin secretion, either directly measured (e.g. c-peptide or HOMA-B) or proxy measures, such as diabetes duration, were associated with reduced glycaemic response to GLP1-RA, although in the majority of evidence was from observational studies. As with other glucose-lowering drug classes, a greater glycaemic response with both SGLT2i and GLP1-RA was seen at higher baseline HbA1c. We did not identify any studies examining whether the relative efficacy of SGLT2i compared to GLP1-RA is altered by baseline HbA1c levels. Of note, many of the included studies for HbA1c outcome were observational, meaning findings could potentially reflect biases from differential prescribing behaviour, or regression to the mean, although we did attempt to account for the latter by including adjustment for baseline HbA1c as one of our study inclusion criteria.*

*For both CVD and heart failure outcomes, RCT meta-analyses do not support differences in the relative efficacy of either GLP1-RA or SGLT2i based on an individuals' prior CVD status. However, this finding should be interpreted cautiously as all RCTs to-date have predominantly included participants with, or at high-risk of, CVD, thereby excluding the majority of the wider T2D population at lower risk. However, meta-analyses suggest **(with moderate certainty) that** the relative effects of both drug classes may be greater in people of non-White ethnicity. In particular, those of Asian and African ethnicity (compared to Whites) have been shown to have a greater relative benefit for hospitalization for heart failure/CV death (but not MACE) with SGLT2i, and MACE for GLP1-RA.”*

Reviewer #2 (Remarks to the Author):

Thanks for the authors' point-to-point responses to my concerns. However, not including the event-driven outcome trials

such as DELIVER, EMPA-Kidney and AMPLITUDE-O is still inappropriate from my own perspective. I am aware that the literature search was conducted in February 2022 when these trials were not officially published. However, it has been more than one year since the primary literature search. The literature search should be updated at this moment. We are talking about the large-scale and well-designed CVOTs instead of certain phase 2 or phase 3 efficacy evaluation trials with limited participants. The authors now are making their efforts to produce a state-of-art summary of high-quality evidence for precision medicine in applying GLP-1RAs and SGLT2i at clinical settings. Without the latest CVOTs included, the analyses could be biased.

>> We appreciate the reviewer's point, and we agree our review represents an evaluation at a (recent) point in time of an evolving field. Our study period (data extraction February 2022) was predetermined and is in-line with the other reviews conducted as part of ADA/EASD Precision Medicine in Diabetes Initiative Consortium, and it is unfortunately unfeasible to update the literature search.

We also agree that the trials highlighted are important trials however all 3 were primary RCTs studies whose question was to determine average treatment effects and not to address treatment effect heterogeneity. Therefore these individual RCTs would not have met our inclusion criteria. It is however possible these trials may provide additional evidence in the future as they could be incorporated into meta-analyses studies that specifically evaluate treatment effect heterogeneity. We have therefore added to the discussion to highlight the time period of our evaluation and the potential that recent informative studies such as these RCTs are not included. This complements our existing paragraph 8 of the Discussion where we have already identified several notable studies published after our evaluation period that have specifically evaluated treatment effect heterogeneity and provided important new information that complements our review. We hope the following additions to the Discussion now more comprehensively highlight this:

Discussion, paragraph 9 (additions in bold): "Of note, several studies published since our data extraction was completed in February 2022 which fill some of the evidence gaps identified in our review, and highlight the clear potential for a precision medicine approach to T2D treatment: the TriMaster study – an 'n-of-1' precision medicine RCT of SGLT2i, DPP4i and thiazolidinediones (TZD) that established that individuals with higher renal function (eGFR >90 ml/min/1.73 m²) have a greater HbA1c response with SGLT2i vs DPP4i relative to those with eGFR 60–90 ml/min/1.73 m² 131, a result concordant with our finding that reduced renal function is associated with lower efficacy of SGLT2i; a similarly designed two-way crossover trial in New Zealand which identified a greater relative benefit of TZD therapy compared to DPP4i in people with obesity and/or hypertriglyceridemia¹³²; a study using large-scale observational data and post-hoc analysis of individual participant-level data from 14 RCTs that specifically investigated differential treatment effects with SGLT2i and DPP4i, and developed a treatment selection model to predict HbA1c response on the two therapies based on an individuals' routine clinical characteristics¹³³; and a robust study across observational and multiple RCTs identifying pharmacogenetic markers of differential glycaemic response to GLP1-RA¹³⁴. In addition, three large trials (AMPLITUDE-O investigating cardiovascular and renal outcomes in 4,076 participants with T2D for the GLP-RA epeglenatide,¹³⁵ DELIVER investigating worsening heart failure or cardiovascular death in 3,131 participants [45% with T2D] for the SGLT2i Dapagliflozin,¹³⁶ and EMPA-KIDNEY investigating progression of kidney disease or cardiovascular death in 6,609 participants [44% with T2D]¹³⁷) have recently been published. Although all three are primary RCTs examining average treatment effects rather than treatment effect heterogeneity, and thus would have been ineligible for our review, future meta-analysis studies integrating the results of these and other ongoing SGLT2i and GLP1-RA trials may add to the evidence we have presented."

Reviewer #3 (Remarks to the Author):

I thank the authors for their comprehensive response to my comments. I remain of opinion that the novelty of this paper is limited and specific gaps to be addressed remain vague. However, i recognize this is subjective and respect the authors have a different opinion.

>> Thank you again for reviewing our study and for the helpful comments.

REVIEWERS' COMMENTS:

Reviewer #1 (Remarks to the Author):

Thanks for the amends from the authors. I am happy to find its publication in the journal.

Reviewer #2 (Remarks to the Author):

I have no further questions.